# Reinforcement Learning based Disease Progression Model for Alzheimer's Disease

**Krishnakant V. Saboo**
UIUC
ksaboo2@illinois.edu

**Anirudh Choudhary**
UIUC
ac67@illinois.edu

**Yurui Cao**
UIUC
yuruic2@illinois.edu

**Gregory A. Worrell**
Mayo Clinic
Worrell.Gregory@mayo.edu

**David T. Jones**
Mayo Clinic
Jones.David@mayo.edu

**Ravishankar K. Iyer**
UIUC
rkiyer@illinois.edu

## Abstract

We model Alzheimer's disease (AD) progression by combining differential equations (DEs) and reinforcement learning (RL) with domain knowledge. DEs provide relationships between some, but not all, factors relevant to AD. We assume that the missing relationships must satisfy general criteria about the working of the brain, for e.g., maximizing cognition while minimizing the cost of supporting cognition. This allows us to extract the missing relationships by using RL to optimize an objective (reward) function that captures the above criteria. We use our model consisting of DEs (as a simulator) and the trained RL agent to predict individualized 10-year AD progression using baseline (year 0) features on synthetic and real data. The model was comparable or better at predicting 10-year cognition trajectories than state-of-the-art learning-based models. Our interpretable model demonstrated, and provided insights into, "recovery/compensatory" processes that mitigate the effect of AD, even though those processes were not explicitly encoded in the model. Our framework combines DEs with RL for modelling AD progression and has broad applicability for understanding other neurological disorders.

## 1 Introduction

Models that describe Alzheimer's disease (AD) progression through time, i.e., the evolution of factors involved in the disease such as brain size, brain activity, pathology, and cognition (Fig. 1), are crucial for mitigating this highly prevalent disease [1, 2]. Such models can enhance our *understanding* of the disease processes and enable crucial applications like *prediction* of long-term cognition trajectories for early detection. Our goal is to develop a model to predict an individual's future AD progression at 1-year intervals based on their baseline (year 0) data. We address this goal by combining differential equations that capture the relationships between some factors, and leveraging reinforcement learning to extract the missing relationships by optimizing a domain knowledge-based reward function.

Differential[1] equation (DE) based models can describe disease progression by expressing domain knowledge as mathematical relationships between different factors [3, 4]. DE-based models have several advantages which make them attractive for disease progression modelling. The interpretability

---

[1]For simplicity, we use the term "differential equation" to denote algebraic and differential equations.

35th Conference on Neural Information Processing Systems (NeurIPS 2021).

of these models can enhance our understanding of disease processes [5, 6]. These models require limited data because the data is only used for parameter estimation. DE-based models' mechanistic nature allows for intervention exploration [6]. However, these models provide an incomplete view of the disease because DEs describing relationships among some of the factors are unavailable [6].

We propose a framework for modeling AD progression that combines differential equations with reinforcement learning (RL) to overcome the above limitation. We assume that the missing relationships are the solution of an optimization problem that can be formulated based on domain knowledge. In our model, this optimization problem's objective function serves as the reward function for reinforcement learning. Therefore, by optimizing the reward, RL extracts the relationships among factors for which explicit DEs are unavailable. Thus, RL combined with the DEs describes the evolution of factors pertinent for modeling AD progression.

In our model, the available DEs define the simulator and the RL agent optimizes the domain-based reward function in the simulator. The parameters of the DEs are based on the available data. We overcome three main challenges for developing the proposed model. The first challenge is to identify the factors involved in each DE. We use domain knowledge to find existing causal relationships between different factors and differential equations relating them (when available).

Second, the DEs relating cognition, brain regions' sizes, and brain activity are unknown (Fig. 1). Since proposing novel DEs relating these factors requires significant scientific knowledge that is still in development, we address this challenge as follows. Multiple brain regions work together to produce cognition [7, 8]. We assume that the working of multiple brain regions is governed by an optimization problem that maximizes cognition while minimizing the cost of cognition [9]. We represent this optimization problem's objective function in terms of cognition, brain size, and brain activity.

Finally, the above optimization problem needs to be solved for multiple time points for modeling disease progression with the solution at time $t$ affecting the solution at time $t' > t$. Therefore, we use RL to optimize the above objective (reward) function.

Our contributions are:

1. We developed an Alzheimer's disease progression model based on domain knowledge that combines DEs and RL. To the best of our knowledge, this is the first attempt at using RL to develop a disease progression model. Our framework is generic and can be used with other DEs, or to provide a basis for modelling other neurological disorders.

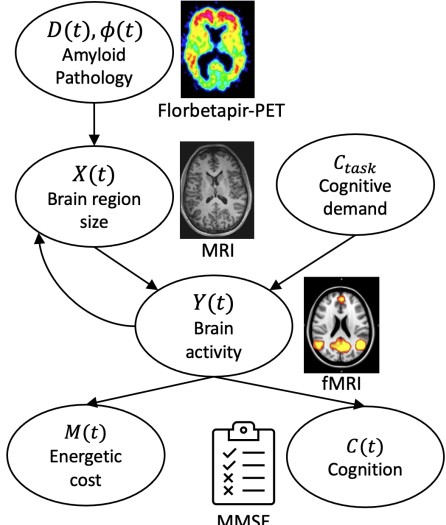

Figure 1: Causal relationships between variables based on domain knowledge.

2. We applied the model for predicting individualized long-term (10-year) future cognition trajectories using baseline ($0^{th}$ year) features on synthetic and real data. On real data, our model reduced the prediction error by $\sim 10\%$ than a state-of-the-art deep learning-based model [10] and produced more realistic trajectories than other benchmark models. Cognition trajectory prediction models are useful in a clinical setting to identify individuals at future risk of cognitive decline.

3. Our interpretable model provided insight into how multiple brain regions together produce cognition during AD. Specifically, we observed "recovery/compensatory" processes that mitigate the effect of AD on cognition [11], which could not be observed with state-of-the-art models. Recovery processes were not explicitly encoded into the model and were an outcome of the reward function's form. Further investigation into the basis of recovery processes could guide the development of interventions.

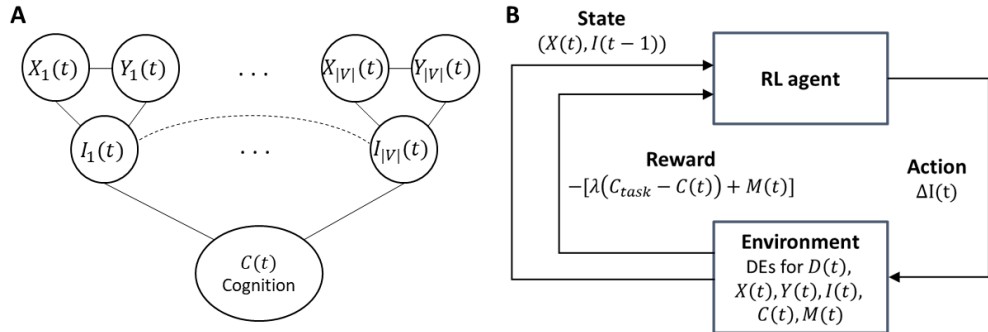

Figure 2: Framework for modeling AD progression. (A) Relationship between brain size, brain activity, information processing, and cognition (represented by solid and dashed edges). (B) Framework for AD progression that combines differential equations (DEs) with reinforcement learning (RL).

## 1.1 Background

Alzheimer's disease results from the interaction of pathology (such as amyloid [12]) and recovery (compensation [13, 11]) processes. These processes jointly affect brain structure (the size of different regions), brain function (activity in different regions), and cognition [14, 12, 11, 15]. Pathology leads to neurodegeneration i.e., reduction in brain size [12, 16], and consequently leads to cognitive decline. Recovery processes compensate for the effect of neurodegeneration on cognition through modification of brain activity by employing other regions of the brain [13] for cognition. Ironically, the compensatory processes can lead to neurodegeneration and accelerated cognitive decline in the long term [15, 11]. An individual's demographics such as gender, education, genetic risk for disease, etc., also play a role in determining disease progression [17].

## 2 Model

First, we use domain knowledge to find existing causal relationships between different factors in Sec 2.1. In Sec 2.2, we define the notation for the factors and substitute their causal relationships with appropriate DEs. In Sec 2.3, we formulate an optimization problem to find missing relationships and solve it using RL. Thus, the model predicts disease progression via the interaction between the DEs (which constitute a simulator) and the action of the RL agent. Finally, we describe the training of the model in Sec. 2.4.

## 2.1 Identification of causal interactions

We simplify the interactions between different factors by using the causal relationships between them proposed in literature (Fig. 1). Amyloid beta (A$\beta$) (measured using florbetapir-PET) is a primary pathological factor in AD. It propagates from one brain region to another through tracts leading to its deposition in neighboring brain regions [3]. Accumulation of amyloid directly affects brain structure (measured using MRI), i.e., brain regions sizes, and leads to neurodegeneration [12]. Brain structure predicts brain activity (measured using fMRI)[18, 19], and brain activity results in cognition (measured using cognitive tests for e.g., MMSE). Since brain activity also depends on cognitive task difficulty [20], we define a hypothetical variable, $C_{task}$, which represents the cognitive demand on the brain and can be thought of as the maximum score on a cognitive test. $C_{task}$ directly influences brain activity. Brain activity has an energetic cost, and itself can lead to neurodegeneration [11]. Developing a model for AD progression is equivalent to describing the differential equation corresponding to each edge in Fig. 1.

## 2.2 Specification of differential equations

**Brain structure:** We represent the brain as the graph $G_S = (V, E)$, where a node $v \in V$ represents a brain region, and an edge $e \in E$ represents a tract. Let $X_v(t)$ denote the size of a brain region $v \in V$ at time $t$, and $X(t) = [X_1(t), X_2(t), ..., X_{|V|}(t)]$.

**Pathology propagation:** To incorporate the propagation and accumulation of pathological A$\beta$ in our framework, we adapt the network diffusion based model proposed by Raj et al. [3, 5] because it captures the propagation of A$\beta$ through tracts. Let $D_v(t)$ be the instantaneous amyloid accumulation in region $v \in V$ at time $t$. Then propagation of amyloid is given by:

$$\frac{dD(t)}{dt} = -\beta H D(t) \tag{1}$$

where, $D(t) = [D_1(t), D_2(t), \ldots, D_{|V|}(t)]$, $H$ is the Laplacian of the adjacency matrix of the graph $G_S$, and $\beta$ is a constant. The total amount of amyloid in a region, $\phi_v(t)$, is:

$$\phi_v(t) = \int_0^t D_v(s) ds \tag{2}$$

**Brain activity and cognition:** Multiple brain regions work in synchrony to produce cognition [8]. Let $Y_v(t)$ denote the activity in region $v \in V$ in support of cognition $C(t)$ at time $t$. Although cognition, brain size ($X_v$), and activity ($Y_v$) are related, the exact relationship among them is unknown and cannot be easily learned from limited data. On the other hand, we can intuitively relate a region's size and activity to its "contribution" to cognition [21, 13]. Therefore, we introduce a hypothetical variable termed as *information processing*, $I_v(t) \in \mathbb{R}_{\geq 0}$, to represent region $v$'s contribution to cognition (Fig. 2). Therefore, the cognition $C(t)$ supported by the brain at time $t$ can be described as:

$$C(t) = \sum_{v \in V} I_v(t). \tag{3}$$

The activity in a given region depends on the amount of information it is processing and its size [21, 13]. For a region $v$ with size $X_v(t)$, the activity $Y_v(t)$ increases as $I_v(t)$ increases [21]. A healthier region (i.e., greater region size) can be considered more efficient and will require lesser activation than an inefficient one for comparable levels of information processing [20]. Therefore, we propose the following relationship between $X_v(t), Y_v(t)$, and $I_v(t)$ with $\gamma$ as a constant:

$$Y_v(t) = \gamma \frac{I_v(t)}{X_v(t)} \quad \forall v \in V \tag{4}$$

$I_v(t)$s are determined through reinforcement learning as described in Sec. 2.3.

**Energetic cost:** The brain consumes energy for supporting cognition. This energy consumption can be thought of as a cost to the brain. The energy consumption in a region is proportional to its activity [22]. Therefore, the energetic cost to the brain is the total brain activity across all the regions. We compute the energetic cost $M(t)$ as follows:

$$M(t) = \sum_{v \in V} Y_v(t) \tag{5}$$

**Degeneration of brain regions:** Neurodegeneration is influenced by two factors: amyloid deposition and brain activity. Previous studies that proposed a linear relationship between the rate of brain region degeneration and A$\beta$ deposition replicated the macroscopic neurodegeneration patterns seen in AD [5, 4]. Moreover, brain activity supporting cognition can further accelerate neurodegeneration [11]. Therefore, we adopted the relationship proposed in [5] and modified it to incorporate the effect of brain activity as follows:

$$\frac{dX_v(t)}{dt} = -\alpha_1 D_v(t) - \alpha_2 Y_v(t) \quad \forall v \in V \tag{6}$$

where $\alpha_1, \alpha_2$ are constants. The first term on the RHS in the Eq. 6 is the degeneration due to A$\beta$ pathology and the second term is the deterioration due to brain activity (see Appendix C.2 for an alternate formulation of this equation).

**Parameter constants of equations:** An individual's demographics, such as gender, genetic risk of AD, education, etc., influence disease progression [17]. In our model, the influence of demographics is mediated through the parameter constants $\alpha_1, \alpha_2, \beta, \gamma$ introduced in Eqs 1, 4, and 6 [5]. For demographic factors $Z_0$ at baseline, let

$$(\alpha_1, \alpha_2, \beta, \gamma) = f(Z_0). \tag{7}$$

## 2.3 Reinforcement learning to determine $I(t)$

Specifying the model requires determining the information processed by each region, i.e., computing $I(t) = [I_1(t), \ldots, I_{|V|}(t)]$. Multiple brain regions jointly produce cognition. Those joint relationships influence $I(t)$. We assume that those joint relationships satisfy general criteria about the working of the brain and obtain $I(t)$ by solving an optimization problem capturing those criteria. Specifically, we assume that $I(t)$ is chosen to distribute the workload related to cognition "optimally" across multiple brain regions. The objective function of that optimization problem is based on two competing criteria: (i) minimizing the deficit between the cognitive demand of a task and the cognition $C(t)$ provided by the brain, and (ii) minimizing the energetic cost $M(t)$ of supporting cognition [9]. The optimization is performed at every time point $t$ with the solution at $t$ affecting the optimization problem at $t' > t$. Therefore, we use RL to determine the optimal $I(t)$ by balancing the two competing criteria. RL is a desirable method in our case since it can train with limited data and also makes the framework generic and flexible to changes/refinements in the reward function and the underlying DEs. Below we describe how RL is adopted in our model (Fig. 2). We implement the model for discrete time steps $t \in \{1, 2, ..., K\}$.

**State and action:** The state $S(t)$ consists of the current size of the brain regions $X(t)$ and the information processed by each region at the previous time point $I(t-1)$. The latter is provided in the state for ease of training the RL model. The action $A(t) \in \mathcal{A}$ specifies the change in information processed by each region from the previous time point, $\Delta I_v(t) \in \mathcal{R} \; \forall v \in V$.

$$S(t) = (X(t), I(t-1)),$$

$$\mathcal{A} = \left\{ \left[ \Delta I_1(t), \ldots, \Delta I_{|V|}(t) \right] \middle| I_v(t) = I_v(t-1) + \Delta I_v(t); \; \sum_{v \in V} I_v(t) \leq C_{task} \right\}$$

**Environment:** The environment is a simulator consisting of the equations relating $D(t), \phi(t), X(t), Y(t), I(t), C(t)$, and $M(t)$ (Eqs. 1, 2, 3, 4, 5, and 6). Based on the action $A(t)$, the environment updates the state and provides a reward to the RL agent.

**Reward:** The policy agent must balance the trade-off between the competing criteria of (i) reducing mismatch between $C_{task}$ and $C(t)$ and (ii) reducing the cost $M(t)$ of supporting cognition. Hence, we define the reward as follows:

$$R(t) = -\left[ \lambda(C_{task} - C(t)) + M(t) \right] \tag{8}$$

where $\lambda$ is a parameter controlling the trade-off between the mismatch and the cost. The goal of the policy agent is to maximize the reward.

## 2.4 Training

Training the model consists of (i) estimating the parameters and finding $f$ in Eq. 7, and (ii) training the RL model (Fig 3). The model is trained using multimodal longitudinal data from individuals in the training set. First, we estimate the parameters $\alpha_1, \alpha_2, \beta, \gamma$ for people in the training set and relate the estimates to baseline demographic features of the individuals $Z_0$ to find $f$. Second, baseline measurements of the $X(0), \phi(0)$ are used with the estimated parameters to train the RL model.

**Parameter estimation:** We assume that for person $i \in \{1, ..., N\}$, data at discrete time points $t \in \{1, ..., K\}$ is available for the variables $X^i(t), D^i(t), Y^i(t), C^i(t)$. We derive parameter estimators by minimizing the L2 norm of the difference between the LHS and RHS of discretized versions of Eqs. 1 for $\beta$, Eq. 6 for $\alpha_1, \alpha_2$, and Eqs. 3 and 4 for $\gamma$ (see Appendix A for the derivations).

Due to unavailability of functional MRI data in our experiments, we also derived parameter estimators for the case in which $Y(t)$ is unavailable for estimation. Minimizing the L2 norm obtained by eliminating $Y_v(t)$ from Eqs. 4, 6, and combining it with Eq. 3 results in the following estimators (see Appendix A for the derivation).

$$\hat{\beta} = -\frac{\sum_i \sum_t (D^i(t))^T H^T \frac{\Delta D^i(t)}{\Delta t}}{\sum_i \sum_t (D^i(t))^T H^T H D^i(t)}, \quad \hat{\alpha_1} = \frac{K_1 K_5 - K_3 K_4}{K_2 K_4 - K_3 K_5}, \quad \hat{\alpha_2 \gamma} = \frac{K_3^2 - K_1 K_2}{K_2 K_4 - K_3 K_5} \tag{9}$$

where, $a_1^i(t) = (X^i(t))^T \frac{\Delta X^i(t)}{\Delta t}, a_2^i(t) = (X^i(t))^T D^i(t)$ are both scalars and $K_1 = \sum_i \sum_t (a_1^i(t))^2, K_2 = \sum_i \sum_t (a_2^i(t))^2, K_3 = \sum_i \sum_t a_1^i(t) a_2^i(t), K_4 = \sum_i \sum_t a_1^i(t) C^i(t), K_5 = \sum_i \sum_t a_2^i(t) C^i(t)$.

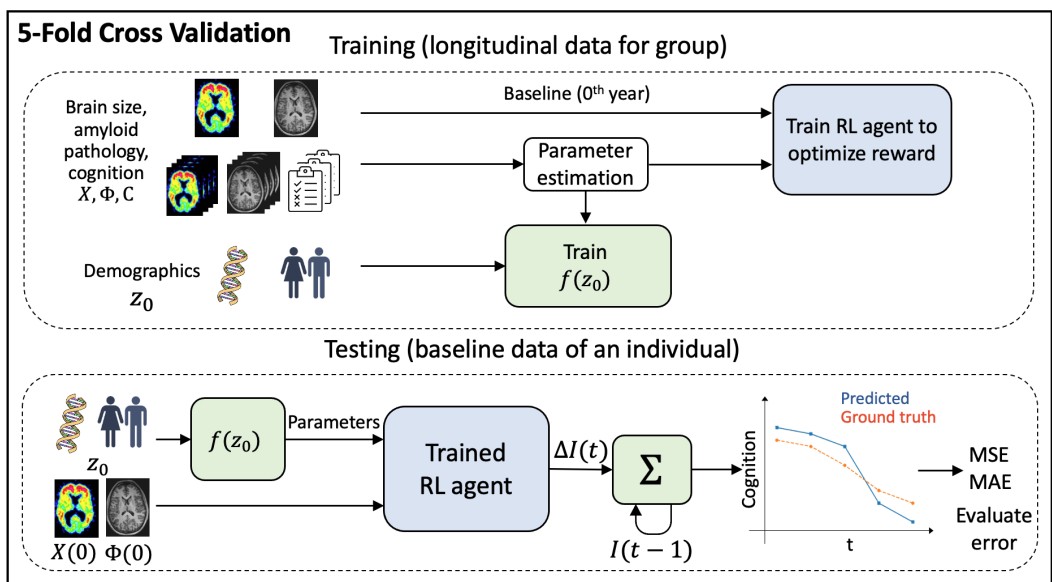

Figure 3: Workflow for training the proposed AD progression model and applying it for long-term cognition trajectory prediction.

Note that in Eq. 9, the product $\alpha_2\hat{\gamma}$ cannot be resolved further due to unavailability of $Y(t)$. Therefore, we assumed that $\gamma$ is the same for all the individuals. Moreover, $\gamma$ affects the model either through $\alpha_2\gamma$ in Eq. 6, or essentially through $\lambda/\gamma$ in the reward function Eq. 8 (see Appendix C.1). Therefore, we set $\gamma = 1$ for all our experiments.

Preliminary analysis showed that only discrete demographic features are associated with the parameter estimates (see Appendix A.6). Therefore, $f$ was the same as a table lookup based on $Z_0$. In a general setting, $f$ can be approximated with a linear model or a neural network.

**RL agent:** The agent is trained on the simulator with model-free on-policy learning using TRPO [23] (see Appendix C.4). All the agent models are trained for 1 million episodes with a batch size of 1000. We clip the reward within a range of [-2000, 2000] with a KL-divergence of 0.01 and generalized advantage estimate discount factor of 0.97. We do not tweak any other hyperparameters of TRPO. The policy network is parameterized by a two-layer feedforward neural network with 32 hidden units each. We implement the simulator using OpenAI's Gym framework [24]. We performed a grid-search for $\lambda$ and $I(0)$ and chose the value that minimized the validation set error (see Appendix C.5).

## 3 Experimental Setup

### 3.1 Adopting AD progression model for 10-year cognition trajectory prediction

During testing, our model can be used for predicting the long-term future cognition trajectory of an individual based on their baseline (0th year) data (Fig. 3). A new person's baseline demographics ($Z_0$) are used to find their parameters $\alpha_1, \alpha_2\gamma, \beta$ using Eq. 7. Those parameters, along with baseline value of brain regions sizes ($X(0)$) and amyloid ($\phi(0)$), are provided as initialization to the trained RL agent for predicting disease progression. We set cognitive demand $C_{task} = 10$ for all the experiments. We predicted cognitive score at 1-year intervals for 10 years after baseline.

### 3.2 Dataset

We validated the model on synthetic data and real-world data derived from the Alzheimer's Disease Neuroimaging Initiative (ADNI) database (adni.loni.usc.edu) [25].

**Synthetic data:** We validated our model on synthetic data of 200 samples. Each sample consisted of longitudinal trajectories of the involved variables (see Appendix B for details).

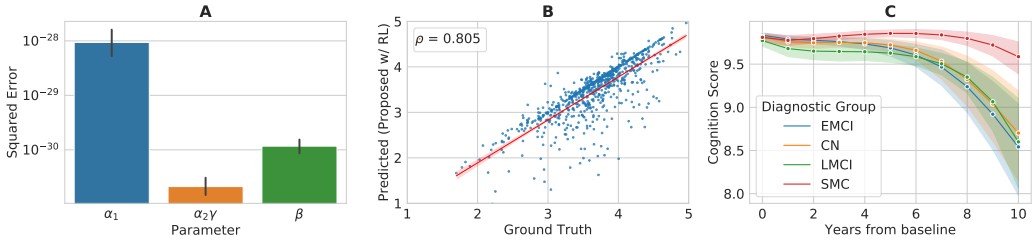

Figure 4: Validation of the proposed model. (A) Error in parameter estimation for synthetic data. Parameters were of the order of $10^{-2}$ to $10^{0}$. (B) Predicted vs. ground truth size of hippocampus for ADNI. (C) Average predicted cognition trajectories over 10 years for groups based on diagnosis at baseline in ADNI.

**ADNI:** In this study, we included data from 160 individuals, used $|V| = 2$ nodes representing the hippocampus (HC) and the prefrontal cortex (PFC), and measured cognition using Mini Mental State Examination (MMSE) (scaled by 3). We excluded fMRI scans from our analysis. See Appendix B for details of data preprocessing.

### 3.3   Model evaluation

Model evaluation consisted of (i) evaluating parameter estimation, (ii) validating group level AD progression trends, (iii) evaluating long-term cognition trajectory prediction, and (iv) providing insights into AD progression. We used 5-fold cross-validation for evaluation with a 64:16:20 split into training, validation, and testing sets. In each fold, each individual's data was only part of one of the three sets. Parameter estimators were evaluated only on the synthetic data using mean squared error, since ground truth values of parameters are known. For evaluating the model's predicted cognition trajectories, we compared them with ground truth trajectories using mean squared error (MSE) and mean absolute error (MAE). Since there are a large number of missing values in ADNI data, we only considered time points for which cognitive scores were available for an individual in computing prediction error. We compared the prediction performance with other state-of-the-art methods (**minimalRNN**, **SVR** [10]; see Appendix D for implementation details) for cognition trajectory prediction. To demonstrate the value of using RL in our framework, we compared the prediction with the case when $I(t)$ is determined by optimizing the reward $R(t)$ using grid-search separately at each time $t$ instead of RL (**w/o RL**) (see Appendix D.3).

## 4   Results

### 4.1   Parameter estimation

We estimated the parameters $\alpha_1, \alpha_2\gamma, \beta$ for the synthetic data using the estimators in Eq. 9. Estimators achieved low errors (Fig. 4). Since real data has a large number of missing values, we also evaluated the effect of missing values on parameter estimation. Although missing values increased the estimation error, the estimated parameters were still comparable to their ground truth values (see Appendix A.5). $Z_0$ consisted of two discrete variables for the synthetic data. Therefore, $f(Z_0)$ (from Eq. 7) was a table lookup for the synthetic data.

For ADNI data, we performed preliminary analysis with individual-specific parameter estimates and chose gender and genetic risk (presence/absence of APOE-e4 genotype) as the demographic variables corresponding to $Z_0$ (see Appendix A.6). We also observed those parameter estimates were highly variable across subjects, which could be due to the small number of data points per individual, missing data, or noise in the data. Therefore, we estimated parameters for groups of subjects based on their gender and genetic risk. The estimated parameters are shown in the Appendix A.6. $f(Z_0)$ was a table lookup with $Z_0 = $ (gender, genetic risk).

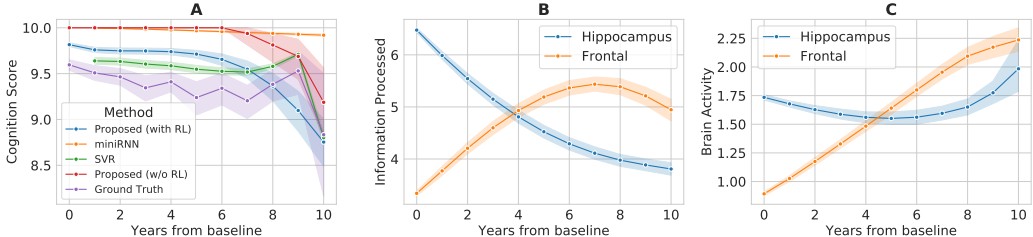

Figure 5: (A) Predicted 10-year future cognition trajectories on the ADNI averaged across individuals. Trajectories are shown for the ground truth, benchmark models, and proposed model. (B) Information processing averaged across individuals for HC (hippocampus) and PFC (frontal) during disease progression for the proposed model. (C) Brain activity averaged across individuals for HC and PFC during disease progression for the proposed model.

## 4.2 Model validation

We validated the proposed model by comparing group-level trends for individuals in ADNI data. First, we compared the size of the brain regions obtained from our model with their size in the ground truth data. The values of regions size obtained from the model were highly correlated with those in the ground truth data (correlation of 0.8 for HC and 0.62 for PFC) (Fig. 4, Fig. 9). We also compared the average cognition trajectories for each diagnostic group based on a person's diagnosis at baseline. As expected, the cognition of early and late mild cognitive impairment (MCI) individuals declined faster than individuals with significant memory concern (SMC) for whom cognitive impairment is less severe in reality (Fig. 4). Counter-intuitively, cognitively normal (CN) individuals showed a rate of decline higher than SMC group and similar to MCI groups. To understand this result, we compared the mean baseline region sizes and amyloid values in different groups. Mean baseline brain region sizes for the CN group were lower than SMC and more similar to MCI groups (mean for HC- CN: 3.81, EMCI: 3.77, LMCI: 3.60, SMC: 3.94; PFC- CN: 3.71, EMCI: 3.81, LMCI: 3.74, SMC: 3.75). Similarly, mean baseline amyloid for CN participants was higher than SMC for HC (CN: 1.32, EMCI: 1.29, LMCI: 1.29, SMC:1.28) although for the frontal areas was smaller than SMC (CN: 1.26, EMCI: 1.30, LMCI: 1.32, SMC: 1.33). Since the disease progression prediction largely depends on the baseline value of the features, these observations taken together suggest that the similarity of baseline features of CN to MCI groups than SMC leads the predicted cognitive trajectories for CN being similar to MCI groups and worse than SMC.

## 4.3 Cognition trajectory prediction

We evaluated the ability of the model to predict individualized 10-year future cognition trajectories based on baseline data (see Table 1). In general, data-driven models tend to be more accurate than mechanistic models and thus provide an upper limit for prediction performance (see Sec. 5). The proposed framework achieved comparable or better performance than state-of-the-art methods on synthetic data as well as real data. The proposed model outperformed the DE-based model without RL, which demonstrates the value of using RL for combining DEs. At an individual level, predicted cognition from the model without RL was either 0 or 10 whereas predictions from the model with RL better resembled ground truth cognition (see Appendix D.3). The improved performance of the proposed model than the state-of-the-art deep learning-based (minimalRNN) model could be due to the small sample size. Although our model had a higher error compared to SVR on ADNI, the average cognition trajectories from our model were more realistic [26] (Fig. 5). Average trajectories for SVR on ADNI showed an increase in cognition at year 9 followed by a steep decline at year 10. This trend could be attributed to the disparity in the number of participants who had data at year 8 (n=60), year 9 (n=17), and year 10 (n=4) with few individuals having cognitive data at year 9 or 10. These observations highlight the value of using DEs and domain knowledge in our model.

We also evaluated the consistency of the trajectories predicted by the proposed model when data from different follow-up times was used as "baseline". 134 participants had all the relevant data available at year 2. For those participants, we predicted trajectories using year 2 as "baseline" and compared

them with the trajectories predicted using year 0 as baseline. The two trajectories for each participant were highly correlated (average correlation across participants=0.93).

| Method | Synthetic Data | | ADNI | |
|---|---|---|---|---|
| | MAE | MSE | MAE | MSE |
| **Proposed (with RL)** | **0.641 (0.090)** | **0.910 (0.229)** | 0.537 (0.127) | 0.761 (0.370) |
| Proposed (w/o RL) | 1.009 (1.670) | 3.806 (8.073) | 0.595 (0.137) | 1.112 (0.406) |
| minimalRNN | 1.395 (0.149) | 6.971 (0.753) | 0.599 (0.137) | 0.984 (0.659) |
| SVR | 0.658 (0.050) | 0.997 (0.112) | **0.495 (0.067)** | **0.574 (0.230)** |

Table 1: Long-term cognition trajectory prediction performance. Standard deviations are provided in parentheses. Abbreviations: MAE, mean absolute error; MSE, mean squared error; and SVR, support vector regression.

## 4.4 Demonstrating recovery

We observed that our model exhibited recovery/compensatory processes during disease progression even though those were not explicitly encoded into the model. On averaging the trajectories of $I_v(t)$ for HC and PFC across individuals, we observed that as the contribution of HC to cognition reduced, the contribution of the PFC increased, resulting in maintaining the total cognition (Fig. 5). As disease progressed further, the information processing in PFC also decreased while the information processing in HC continued to decrease, resulting in cognitive decline. This observation is consistent with recovery/compensatory behaviour that has been hypothesized to explain the response of the brain to injury [13, 11]. In control experiments, we observed that the proposed model trained with a modified reward function that only included the cognition mismatch term or the energetic cost term (in Eq. 8) did not show recovery processes (see Appendix D.4). Our observation of recovery processes emerging from the model formulation, specifically the reward function, is crucial because it could provide further understanding of the basis of those processes. Recovery-like behavior has also been seen in other neurological disorders [11], and understanding the mechanisms driving them could be useful for exploring interventions for AD. Our model also showed an initial phase of increased brain activity (hypermetabolism) in the PFC which is consistent with previous observations in AD and aging studies [13, 27]. The interpretable nature of our model enables these observations, which can help us better understand AD. Note that the interpretation of these results is meaningful within the context of the assumptions of the model and further research is needed to validate them.

## 5 Related Work

Studies on modelling Alzheimer's disease progression can be broadly categorized as follows:

1. *Data-driven models* fit parametric/non-parametric models on biomarker data to relate disease pathology, region size, activity, cognition, and/or demographics. Examples of these models include event-based models [28], mixed-effects models [29], Bayesian models [30], and machine learning-based approaches [10, 31, 32, 33]. These methods have a low dependency on domain knowledge and work well for short-term prediction, but their long-term prediction performance is limited [34] due to lack of longitudinal multimodal data spanning the entire disease course. AD progresses over decades [35, 36], so accurate modeling requires long-term longitudinal data from multiple modalities (MRI, PET, cognitive assessments) [35], which is difficult to acquire. Further, interpretability of these models may be a challenge depending on their complexity which limits their applicability in understanding disease processes.

2. *Mechanistic models* use domain knowledge to represent the relationship among variables using algebraic and/or differential equations [4, 3]. Popular approaches include graph-based models [37, 38] and dynamic causal modeling [39]. These models are interpretable and have a low dependence on data. Although studies have modelled the relationships among a subset of the variables relevant for AD progression, to the best of our knowledge, a model that brings together pathology, brain structure, function, cognition, and demographics is yet to be proposed.

Our model enjoys the advantages offered by mechanistic models through the use of several DEs in its formulation. It differs from the existing mechanistic models by incorporating the relationship between brain size, activity, and cognition. Moreover, the use of an RL model trained in a DE-based simulator using a domain-guided reward function allows leveraging recent advances in machine learning without increasing the model's dependence on data.

Reinforcement learning has been used previously for developing treatment paradigms for neurological disorders such as epilepsy [40] and Parkinson's disease [41]. Recently, Krylov et al. [41] showcased the efficacy of a policy-based RL framework in suppressing neuronal synchrony in Parkinson's disease. They incorporated DE-based neuronal models into a simulator and trained multiple PPO agents for oscillatory neuronal models. We are unaware of previous work that has used RL for AD disease progression modelling.

## 6 Conclusion and Future Work

We proposed a framework for modeling AD progression by combining differential equations and reinforcement learning and evaluated the validity of its predictions on real and synthetic data. Our model successfully predicted individualized 10-year-long future cognition trajectories, which can be useful in a clinical setting for identifying individuals at a future risk of decline. Our model demonstrated, and provided insights into the potential basis of, recovery processes during AD progression which emerged from the model formulation, specifically the reward function. Thus, the model can help in further understanding disease progression. The mechanistic nature of our model allows the exploration of interventions strategies through perturbation analysis of the variables [6]. We contend that our framework is generic and can be modified to support other DEs describing AD, or provide a basis for modelling other neurological disorders because of the pervasiveness of recovery processes [11] and the prevalence of using differential equations for modelling diseases [42, 43].

There are some limitations of the current work. We demonstrated the model with a 2-node graph, which is too coarse to capture the multimodal changes happening at finer spatial scales in the brain's structure and activity during AD progression. The model can scale to larger graphs with physiologically relevant nodes using data from the relevant brain regions. For relating brain activity to size, we proposed an inverse linear model (Eq. 4), but due to lack of sufficient fMRI data in our experiments, we could not validate that equation. Further experiments with fMRI data may lead to a better model for relating brain size ($X(t)$), activity ($Y(t)$), and information processing ($I(t)$). AD also results in tract degradation which we have not modeled. Related to this is the sensitivity of the model to misspecification of the dynamical system. We observed that the form of the DEs affects disease progression prediction (see Appendix C.3) but the extent of the effect and its dependence on different factors needs further clarification. Finally, the model was only validated on ADNI data and, therefore, suffers from the same biases inherent in the data [25]. Further experiments with larger and diverse data is needed to validate the results of this study. We plan to address these limitations in future work.

### Acknowledgments and Disclosure of Funding

This work was supported by the Mayo Clinic and Illinois Alliance Fellowship for Technology-based Healthcare Research and in part by NSF grants CNS-1337732, CNS-1624790, and CCF-2029049 and the Jump ARCHES endowment fund. We thank Saurabh Jha, Subho Banerjee, Frances Rigberg, and Prakruthi Burra for their valuable feedback.

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
