# Appendix

## A    Parameter estimators derivation and analyses

Parameters $\alpha_1, \alpha_2, \beta, \gamma$ introduced in Eqs. 1, 6, 4 are crucial to replicate the AD progression observed in real data. However, the appropriate values of the parameter are not known *a priori* and have to be estimated from the data. Below, we derive optimal estimators for each of the parameters. For the following derivations, we assumed that longitudinal features $X^i(t), Y^i(t), \phi^i(t), D^i(t), C^i(t) \; \forall \, t \in \{0, 1, 2..., K\}$ are available for individual $i \in \{1, ..., N\}$. Estimators are derived for a general setting of "same parameter for a group of individuals" but the estimators are also applicable per-individual.

### A.1    Estimation of $\gamma$

Parameter $\gamma$ in Eq. 4 relates activity, size, and information processed by a region. Since $I(t)$ is not measured physically, we rearranged the equation to express it in terms of $\gamma, Y_v^i(t), X_v^i(t)$. We then used Eq. 3 and setup an L2 minimization problem on the available data.

$$\mathcal{L}\left(\frac{1}{\gamma}\right) = \sum_i \sum_t \left( C^i(t) - \sum_{v \in V} \frac{1}{\gamma} Y_v^i(t) X_v^i(t) \right)^2$$

$$\gamma^* = \arg\min_{\gamma \in \mathcal{R}} \mathcal{L}\left(\frac{1}{\gamma}\right)$$

Replacing $\psi = \frac{1}{\gamma}$, and setting derivative to zero, we get:

$$\frac{d\mathcal{L}(\psi)}{d\psi} = -\sum_i \sum_t 2 \left( C^i(t) - \psi \sum_{v \in V} Y_v^i(t) X_v^i(t) \right) \sum_{v \in V} Y_v^i(t) X_v^i(t)$$

$$0 = \sum_i \sum_t C^i(t)(Y^i(t))^T X(t) - \psi \sum_i \sum_t ((Y^i(t))^T X^i(t))^2$$

$$\psi = \frac{\sum_i \sum_t C^i(t)(Y^i(t))^T X^i(t)}{\sum_i \sum_t ((Y^i(t))^T X^i(t))^2}$$

### A.2    Estimator for $\beta$

$\beta$ can be estimated by setting up an L2 optimization problem using Eq. 1. $D^i(t)$ and $H$ are available from measurements. We approximate $\frac{dD_v(t)}{dt} \approx \frac{\Delta D_v(t)}{\Delta t} \; \forall v \in V$, the RHS of which is available from longitudinal measurements of $D(t)$.

$$\mathcal{L}(\beta) = \sum_i \sum_t \left\| \frac{\Delta D^i(t)}{\Delta t} + \beta H D^i(t) \right\|_2^2 = \sum_i \sum_t \left( \frac{\Delta D^i(t)}{\Delta t} + \beta H D^i(t) \right)^T \left( \frac{\Delta D^i(t)}{\Delta t} + \beta H D^i(t) \right)$$

$$\beta^* = \arg\min_{\beta \in \mathcal{R}} \mathcal{L}(\beta)$$

Setting derivative wrt $\beta$ to zero, we get:

$$\frac{d\mathcal{L}(\beta)}{d\beta} = 2 \sum_i \sum_t (H D^i(t))^T \left( \frac{\Delta D^i(t)}{\Delta t} + \beta H D^i(t) \right)$$

$$0 = \sum_i \sum_t (D^i(t))^T H^T \frac{\Delta D^i(t)}{\Delta t} + \beta \sum_i \sum_t (D^i(t))^T H^T H D^i(t)$$

$$\beta = -\frac{\sum_i \sum_t (D^i(t))^T H^T \frac{\Delta D^i(t)}{\Delta t}}{\sum_i \sum_t (D^i(t))^T H^T H D^i(t)}$$

## A.3   Estimators for $\alpha_1, \alpha_2$

$\alpha_1$ and $\alpha_2$ can be estimated using L2 norm optimization and Eq. 6. Rearranging terms in Eq. 6, we get:

$$\frac{\Delta X_v^i(t)}{\Delta t} = -[D_v^i(t) \ Y_v^i(t)] \begin{bmatrix} \alpha_1 \\ \alpha_2 \end{bmatrix} \quad \forall v \in V$$

Stacking all the nodes together gives the following matrix notations, with $Q(t) = [D^i(t)Y^i(t)]$ is a matrix with $D(t)$ and $Y(t)$ as its columns, and $\alpha = [\alpha_1 \ \alpha_2]^T$,

$$\frac{\Delta X^i(t)}{\Delta t} = -Q^i(t)\alpha$$

The optimization problem is as follows:

$$\mathcal{L}(\alpha) = \sum_i \sum_t \left\| \frac{\Delta X^i(t)}{\Delta t} + Q^i(t)\alpha \right\|_2^2 = \sum_i \sum_t \left( \frac{\Delta X^i(t)}{\Delta t} + Q^i(t)\alpha \right)^T \left( \frac{\Delta X^i(t)}{\Delta t} + Q^i(t)\alpha \right)$$

$$\alpha^* = \arg \min_{\alpha \in \mathcal{R}^2} \mathcal{L}(\alpha)$$

Setting the derivative to 0 and simplifying, we get:

$$\nabla \mathcal{L}(\alpha) = \sum_i \sum_t 2(Q^i(t))^T \left( \frac{\Delta X^i(t)}{\Delta t} + Q^i(t)\alpha \right)$$

$$0 = \sum_i \sum_t (Q^i(t))^T \frac{\Delta X^i(t)}{\Delta t} + \sum_i \sum_t (Q^i(t))^T Q^i(t)\alpha$$

$$\alpha = -\left( \sum_i \sum_t (Q^i(t))^T Q^i(t) \right)^{-1} \left( \sum_i \sum_t (Q^i(t))^T \frac{\Delta X^i(t)}{\Delta t} \right)$$

## A.4   Estimating parameters when $Y(t)$ is unavailable

New parameter estimators that leverage only the available data need to be derived when $Y(t)$ is unavailable. The derivation goes as follows: first, we eliminate $Y(t)$ from the model equations. Second, we set up an L2 optimization problem involving the parameters and measured variables in the updated equation. Finally, we optimize the L2 objection function to derive optimal parameters.

Substituting Eq. 4 in Eq. 6 and expressing $I_v(t)$ in terms of the remaining variables, we get:

$$\frac{dX_v(t)}{dt} = -\alpha_1 D_v(t) - \alpha_2 \gamma \frac{I_v(t)}{X_v(t)}$$

$$X_v(t)\frac{dX_v(t)}{dt} = -\alpha_1 X_v(t)D_v(t) - \alpha_2 \gamma I_v(t)$$

$$I_v(t) = -\frac{1}{\alpha_2 \gamma} \left( X_v(t)\frac{dX_v(t)}{dt} + \alpha_1 X_v(t)D_v(t) \right)$$

Substituting the above expression in Eq. 3, denoting $\theta = (\alpha_1, \alpha_2\gamma)$, and setting up an L2 optimization problem based on it, we get:

$$\mathcal{L}(\theta) = \sum_{i,t} \left[ C^i(t) + \frac{1}{\alpha_2 \gamma} \sum_{v \in V} \left( X_v^i(t)\frac{dX_v^i(t)}{dt} + \alpha_1 X_v^i(t)D_v^i(t) \right) \right]^2$$

We approximate $\frac{dX_v^i(t)}{dt} \approx \frac{\Delta X_v^i(t)}{\Delta t}$. Additionally, $\sum_{v \in V} X_v^i(t)\frac{\Delta X_v^i(t)}{\Delta t} = (X^i(t))^T \frac{\Delta X^i(t)}{\Delta t}$ Similarly, $\sum_{v \in V} X_v^i(t)D_v^i(t) = (X^i(t))^T D^i(t)$. This gives:

$$\mathcal{L}(\theta) = \sum_{i,t} \left[ C^i(t) + \frac{1}{\alpha_2 \gamma} \left( (X^i(t))^T \frac{\Delta X^i(t)}{\Delta t} + \alpha_1 (X^i(t))^T D^i(t) \right) \right]^2$$

To simplify notation, we replace $\delta_1 = \frac{1}{\alpha_2 \gamma}$, $\delta_2 = \alpha_1$, $a_1^i(t) = (X^i(t))^T \frac{\Delta X^i(t)}{\Delta t}$, and $a_2^i(t) = (X^i(t))^T D^i(t)$.

$$\mathcal{L}(\theta) = \sum_{i,t}(C^i(t) + \delta_1 a_1^i(t) + \delta_1 \delta_2 a_2^i(t))^2$$

$$\theta^* = \arg \min_{\delta_1, \delta_2 \in \mathcal{R}} \mathcal{L}(\theta)$$

$\delta_1$ and $\delta_2$ are coupled in the above optimization problem. We compute the partial derivative with respect to each variable and simply the resulting set of equations. First, computing the derivative with respect to $\delta_2$.

$$\frac{\partial \mathcal{L}(\theta)}{\partial \delta_2} = \sum_{i,t} 2(C^i(t) + \delta_1 a_1^i(t) + \delta_1 \delta_2 a_2^i(t))(0 + 0 + \delta_1 a_2(t))$$

$$0 = 2\delta_1 \sum_{i,t} a_2^i(t)(C^i(t) + \delta_1 a_1^i(t) + \delta_1 \delta_2 a_2^i(t))$$

$$0 = \sum_{i,t} a_2^i(t)C(t) + \delta_1 a_2^i(t)a_1^i(t) + \delta_1 \delta_2 (a_2^i(t))^2$$

$$\therefore \quad -\sum_{i,t} a_2(t)C(t) = \delta_1 \sum_{i,t} a_2^i(t)a_1^i(t) + \delta_1 \delta_2 \sum_{i,t} (a_2^i(t))^2$$

$$\therefore \quad \delta_1 = -\frac{\sum_{i,t} a_2^i(t)C^i(t)}{\sum_{i,t} a_2^i(t)a_1^i(t) + \delta_2 \sum_{i,t}(a_2^i(t))^2} \tag{10}$$

Now, computing partial derivative wrt $\delta_1$ and substituting its values from the Eq. 10, we get

$$\frac{\partial \mathcal{L}(\theta)}{\partial \delta_1} = \sum_{i,t} 2(C^i(t) + \delta_1 a_1^i(t) + \delta_1 \delta_2 a_2^i(t))(a_1^i(t) + \delta_2 a_2^i(t))$$

$$\therefore \quad 0 = \sum_{i,t} C^i(t)(a_1^i(t) + \delta_2 a_2^i(t)) + \delta_1 (a_1^i(t) + \delta_2 a_2^i(t))^2$$

$$\therefore \quad \sum_{i,t} C^i(t)(a_1^i(t) + \delta_2 a_2^i(t)) = -\sum_{i,t} \delta_1 (a_1^i(t) + \delta_2 a_2^i(t))^2$$

$$\therefore \quad \sum_{i,t} C^i(t)a_1^i(t) + \delta_2 \sum_{i,t} C^i(t)a_2^i(t) = -\sum_{i,t} \delta_1 (a_1^i(t) + \delta_2 a_2^i(t))^2 \tag{11}$$

Considering the RHS in the above equation separately for simplification, we get,

$$RHS = -\sum_{i,t} \delta_1((a_1^i(t))^2 + 2\delta_2 a_1^i(t)a_2^i(t) + \delta_2^2(a_2^i(t))^2)$$

$$= -\delta_1 \left( \sum_{i,t} a_1^2(t) + 2\delta_2 \sum_{i,t} a_1(t)a_2(t) + \delta_2^2 \sum_{i,t} a_2^2(t) \right)$$

We make the following substitutions for ease of notation: $K_1 = \sum_{i,t}(a_1^i(t))^2$, $K_2 = \sum_{i,t}(a_2^i(t))^2$, $K_3 = \sum_{i,t} a_1^i(t)a_2^i(t)$, $K_4 = \sum_{i,t} a_1^i(t)C^i(t)$, and $K_5 = \sum_{i,t} a_2^i(t)C^i(t)$. LHS and RHS of Eq. 11 can be re-written with the above constants as follows:

$$LHS = \sum_{i,t} C^i(t)a_1^i(t) + \delta_2 \sum_{i,t} C^i(t)a_2^i(t)$$

$$= K_4 + \delta_2 K_5$$

$$RHS = -\delta_1 \left( \sum_{i,t} a_1^2(t) + 2\delta_2 \sum_{i,t} a_1(t)a_2(t) + \delta_2^2 \sum_{i,t} a_2^2(t) \right)$$

$$= -\delta_1(K_1 + 2\delta_2 K_3 + \delta_2^2 K_2)$$

$$= \left( \frac{K_5}{K_3 + \delta_2 K_2} \right)(K_1 + 2\delta_2 K_3 + \delta_2^2 K_2)$$

We equate LHS=RHS. That gives:

$$(K_4 + \delta_2 K_5)(K_3 + \delta_2 K_2) = K_5(K_1 + 2\delta_2 K_3 + \delta_2^2 K_2)$$

$$\therefore \quad K_3 K_4 + \delta_2 K_2 K_4 + \delta_2 K_3 K_5 + \delta_2^2 K_2 K_5 = K_1 K_5 + 2\delta_2 K_3 K_5 + \delta_2^2 K_2 K_5$$

$$\therefore \quad K_3 K_4 + \delta_2 K_2 K_4 + \delta_2 K_3 K_5 = K_1 K_5 + 2\delta_2 K_3 K_5$$

$$\therefore \quad \delta_2 K_2 K_4 + \delta_2 K_3 K_5 - 2\delta_2 K_3 K_5 = K_1 K_5 - K_3 K_4$$

$$\therefore \quad \delta_2(K_2 K_4 + K_3 K_5 - 2K_3 K_5) = K_1 K_5 - K_3 K_4$$

$$\delta_2 = \frac{K_1 K_5 - K_3 K_4}{K_2 K_4 - K_3 K_5}$$

Substituting $\delta_2$ back in Eq. 10 and simplifying, we get:

$$\delta_1 = \frac{K_2 K_4 - K_3 K_5}{K_3^2 - K_1 K_2}$$

Rewriting in terms of the parameters,

$$\alpha_1 = \frac{K_1 K_5 - K_3 K_4}{K_2 K_4 - K_3 K_5}$$

$$\frac{1}{\alpha_2 \gamma} = \frac{K_2 K_4 - K_3 K_5}{K_3^2 - K_1 K_2}$$

### A.5 Effect of missing values on parameter estimation

In the ADNI data in this study, data is available only for $31\%$ of all possible visits (11 in total) for an individual on average – there are a large number of missing values. To assess the effect of missing values on parameter estimation, we performed parameter estimation on the synthetic data by artificially removing measurements from its samples' trajectories. In the synthetic data, visits from a given year were randomly removed across the population so as to match the percentage of samples missing for that visit year in the real data. Data was removed in this way to ensure that the follow-up years data availability in the synthetic data matched the real data. On this synthetic data with missing values, we estimated the parameters for each individual separately and for a groups of individuals based on their demographics. The squared error of the estimated parameters are shown in Fig. 6. Al-

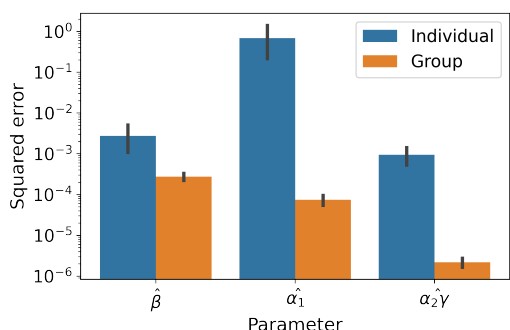

Figure 6: Effect of missing values on parameter estimation. Analysis was performed for the synthetic data.

though the estimation error was higher compared to the case when all the data was available (Fig. 4), estimating parameters for group of individuals achieved lower error than individualized estimation. The ground truth parameter values are of the order of $10^{-2}$ to $10^0$. Therefore, estimation errors for groups are tolerable.

### A.6 ADNI - Selection of demographic variables and parameter estimates

To identify the demographic variables $Z_0$ and their relation to parameters, $f(Z_0)$ (Eq. 7), we followed a two step procedure. First, we estimated the parameters separately for each individual. Second, we performed statistical analysis to find associations between the estimated parameters and the demographic variables. The distribution of individualized parameter estimates are shown in Figure 7. During statistical analysis, we tested for association between each parameter and demographic variable pair. For discrete demographic variables, statistical association was tested with a Wilcoxon ranksum test. For continuous demographic variables, we fit a line with the demographic variable as the independent and the parameter as the dependent variable, and evaluated the p-value of the slope. Baseline age, education, APOE-$\epsilon$4 genotype, and gender were the demographic variables we

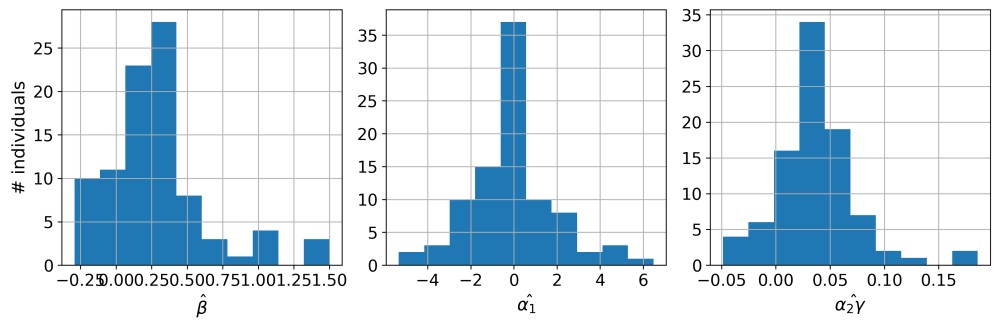

Figure 7: Individualized parameter estimation for split #3 in ADNI. For visual clarity, only values between 5-95%ile are plotted for each parameter.

considered. Parameter estimation and statistical analysis was done separately for each training split in the 5-fold cross-validation. The results of the statistical analysis are shown in Table 2. Gender and genetic risk are the only demographic variables that are associated with some parameter in at least one split. Therefore, we chose those two to represent $Z_0$.

| Parameter | Demographic ft. | $p$-value | | | | |
|---|---|---|---|---|---|---|
| | | Split 0 | Split 1 | Split 2 | Split 3 | Split 4 |
| $\hat{\beta}$ | Age | 0.472 | 0.669 | 0.796 | 0.79 | 0.557 |
| $\hat{\beta}$ | Education | 0.589 | 0.61 | 0.591 | 0.378 | 0.947 |
| $\hat{\beta}$ | Gender | 0.381 | 0.802 | 0.708 | 0.766 | 0.879 |
| $\hat{\beta}$ | **APOE-$\epsilon$4** | **0.003** | 0.312 | 0.402 | **0.026** | 0.076 |
| $\hat{\alpha_1}$ | AGE | 0.224 | 0.47 | 0.331 | 0.499 | 0.238 |
| $\hat{\alpha_1}$ | Education | 0.227 | 0.915 | 0.331 | 0.114 | 0.951 |
| $\hat{\alpha_1}$ | Gender | 0.583 | 0.359 | 0.392 | 0.373 | 0.937 |
| $\hat{\alpha_1}$ | APOE-$\epsilon$4 | 0.596 | 0.316 | 0.523 | 0.115 | 0.197 |
| $\hat{\alpha_2}\gamma$ | Age | 0.799 | 0.798 | 0.253 | 0.77 | 0.718 |
| $\hat{\alpha_2}\gamma$ | Education | 0.904 | 0.133 | 0.821 | 0.244 | 0.079 |
| $\hat{\alpha_2}\gamma$ | **Gender** | 0.276 | 0.102 | **0.049** | **0.0005** | 0.056 |
| $\hat{\alpha_2}\gamma$ | APOE-$\epsilon$4 | 0.884 | 0.492 | 0.357 | 0.234 | 0.915 |

Table 2: Associations between individualized parameter estimates and demographic features. Cases where $p < 0.05$ are highlighted.

There are a large number of missing values in the ADNI data in our study. Estimating parameters based on groups mitigates the effect of missing values. Therefore, we estimated parameters for groups based on the gender and APOE-$\epsilon$4 status. This resulted in 4 groups. Parameter estimates for groups for ADNI data are shown in Figure 8.

## B  Data generation and pre-processing

### B.1  Synthetic data

We generated synthetic data of 200 samples (individuals) to validate the model's ability to model AD progression and predict long-term cognition trajectories. We set $|V| = 2$ and generated longitudinal trajectories of $X(t), Y(t), D(t), \phi(t), I(t)$, and $C(t)$ for $K = 10$ time points (+baseline). The baseline value of brain region size $X(0)$ and amyloid $D(0)$ were generated randomly; $X(0) \sim \mathcal{N}(\mu = [3.5, 3.4], \Sigma = [[0.49, 0.20], [0.20, 0.64]])$ and $D_1(0), D_2(0) \sim Uniform(0, 0.2)$. Each individual also had two discrete demographic features, which were randomly sampled with equal probability from four and two possible values, respectively. This resulted in eight groups of individuals based on their demographic features. The parameters $\alpha_1, \alpha_2, \beta$ were computed from a predetermined linear combination of the demographic features. We set $\gamma = 1$ to be the same

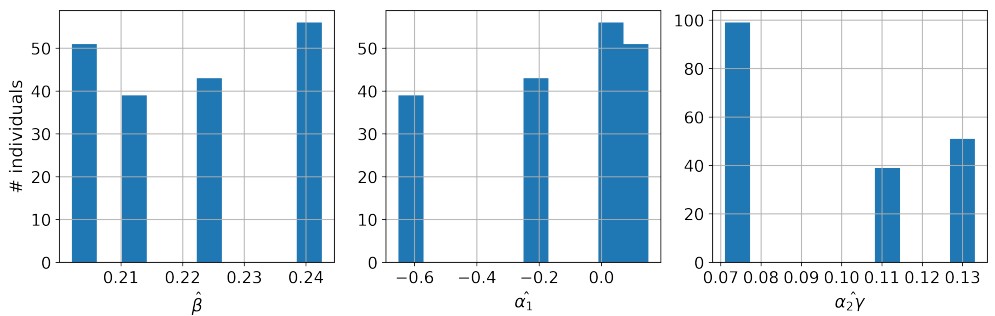

Figure 8: Group-wise parameter estimation for split #3 in ADNI.

for all the individuals and $C_{task} = 10$. Finally, we set $I_1(t) = \min(Y_{max}X_1(t)/\gamma, C_{task})$ and $I_2(t) = \min(Y_{max}X_2(t)/\gamma, C_{task} - I_1(t))$ where $Y_{max} = 2.5$. Based on the above and Eqs. 1, 2, 3, 4, 6, we generated the longitudinal trajectories of $C(t), X(t), D(t), \phi(t), Y(t), I(t)$ for $t \in \{1, 2, .., 15\}$ and stored the last 11 time points to get trajectories that were heterogeneous.

### B.2    ADNI data

We used the ADNI dataset to evaluate the model on real-world data for Alzheimer's disease. For this study, we only included individuals who had (i) baseline ($0^{th}$ year) measurements of cognition, demographics, MRI, and florbetapir PET scans; (ii) longitudinal measurements of cognition; and (iii) at least 2 follow-up measurements (after baseline) that contain both PET and MRI scans along with cognitive assessment at those visits. Visits were not required to be successive, and only 10 years of assessments after baseline were retained for individuals with longer follow-ups. Note that cognitive assessments were retained for all available points upto and including year 10 irrespective of the availability of MRI/PET from those visits. These constraints were chosen to have sufficient measurements for (per-individual) parameter estimation. This resulted in data from 160 participants out of which 52 were cognitively normal (CN), 23 had significant memory concern (SMC), 58 had early mild cognitive impairment (EMCI), and 27 were diagnosed with late MCI (LMCI).

Age, gender, education, and presence of APOE-$\epsilon 4$ genotype were the demographic features. We considered a $|V| = 2$ node graph $G_S$ with nodes representing the hippocampus (HC) and the prefrontal cortex (PFC) due to their importance in supporting cognition and their role in AD [12, 11, 13]. AD pathology primarily targets the hippocampus, a region that plays an important role in memory and cognition, and propagates to other areas of the brain. PFC is involved in executive function in healthy individuals and shows increased activation in older adults and AD patients during cognitive tasks [13]. Volume of the hippocampus (HC) and prefrontal cortex (PFC) were used to represent brain structure ($X(t)$). Raw values of hippocampal volume were divided by $2 \times 10^3$ to be close to 3. To account for the larger size of the PFC in the brain compared to the hippocampus, PFC volumes were normalized by the median ratio of PFC to hippocampus $\times 2 \times 10^3$. Median ratio was computed only on the training data. PET-scan derived SUVR values for PFC and hippocampus were used as a measure of A$\beta$ deposition ($\phi(t)$). Pre-processed files for MRI and PET available on LONI website were used. We used the score on the Mini Mental State Examination (MMSE) for as a measure of cognition. We scaled MMSE score by 3 such that 10 represented perfect cognition and 0 represented no cognition so as to match the choice of $C_{task} = 10$. Although ADNI also contains functional MRI scans, we did not include them in our analysis, since very few individuals had that information along with the rest of the variables.

## C    Model design choices and training

### C.1    Effect of $\gamma$ on the model

$\gamma$ parameter appears at two places in our model. In Eq. 6 through $Y(t)$ and in the reward function Eq. 8. From the parameter estimators in Eq. 9, it is clear that the effect of $\gamma$ on brain size occurs through

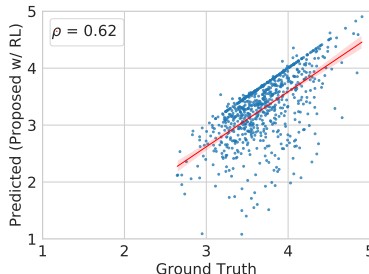

Figure 9: Predicted (proposed model with RL) vs ground truth size of PFC for ADNI data.

the term $\alpha_2\gamma$ which can be estimated from the data. We can rewrite the reward function as:

$$R(t) = -\left[\lambda(C_{task} - C(t)) + M(t)\right]$$

$$= -\left[\lambda(C_{task} - \sum_{v \in V} I_v(t)) + \sum_{v \in V} \frac{\gamma I_v(t)}{X_v(t)}\right]$$

$$= -\gamma\left[\frac{\lambda}{\gamma}(C_{task} - \sum_{v \in V} I_v(t)) + \sum_{v \in V} \frac{I_v(t)}{X_v(t)}\right]$$

Since the optimization of the reward is performed over $I(t)$, $\gamma$ influences the optimization only through $\lambda/\gamma$. Therefore, we set $\gamma = 1$ and varied $\lambda$ in our experiments.

For a similar reason, we modeled $M(t) = \sum_{v \in V} Y_v(t)$ instead of adding a proportionality constant. If there were a proportionality constant $\mu$ such that $M(t) = \mu \sum_{v \in V} Y_v(t)$, the effect of $\mu$ on the overall optimization would only occur through the term $\lambda/\mu\gamma$.

### C.2 Brain activity related degeneration

Although the effect of brain activity on neurodegeneration could be mediated through amyloid [44, 11, 15], we have included activity-based degeneration as a separate term in Eq. 6. This allows us to account for other pathways via which brain activity could lead to atrophy, e.g., oxidative stress and tauopathy. We also developed an alternate version of the proposed model that included an activity-related amyloid deposition term in Eq. 1 instead of the term in Eq. 6:

$$\frac{dD(t)}{dt} = -\beta_1 HD(T) + \beta_2 Y(t); \quad \frac{dX_v(t)}{dt} = -\alpha D(t)$$

The model with the above equations could not be trained on real data due to insufficient number of followups for parameter estimation of an individual. Due to the unavailability of $Y(t)$ and the second order dependence of $\frac{dD(t)}{dt}$ on $\phi(t)$, parameter estimators for $\beta_1, \beta_2$ required at least 4 measurements of $X(t), \phi(t), C(t)$ for estimation. Majority of the individuals (>100) in our data only had 3 measurements of $X(t), \phi(t)$. Nevertheless, we plan to appropriately include the effect of brain activity on degeneration via amyloid in future work.

### C.3 Modified relationship between $Y(t), X(t), I(t)$

In Eq 4, we used an inverse relationship between $Y(t)$ and $X(t)$ to capture the increase in activity for reduced region sizes. To assess the model's sensitivity to the form of the equations, we replaced Eq 4 with a squared inverse relationship between $Y(t)$ and $X(t)$, as follows:

$$Y_v(t) = \gamma \frac{I_v(t)}{X_v^2(t)} \quad \forall v \in V. \tag{12}$$

For this modified model, we derived the parameter estimators following the procedure in Appendix A; computed the parameters for each participant and group of participants; trained the RL agent with Eq 12 in the simulator instead of Eq 4; and evaluated cognition trajectory prediction performance

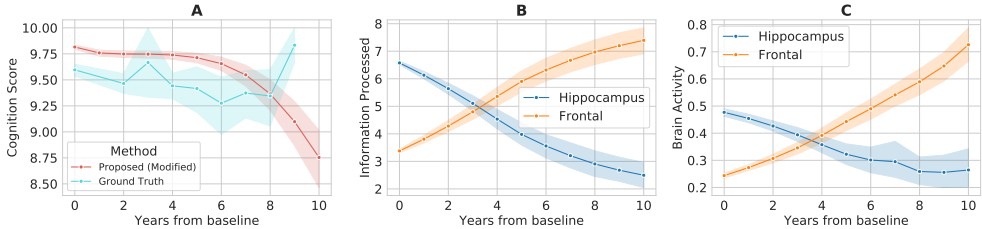

Figure 10: Modified model with inverse squared relationship between $Y(t)$ and $X(t)$ applied to ADNI data. (A) Cognition trajectories from ground truth and model. (B) Information processed in HC (Hippocampus) and PFC (Frontal) averaged across individuals. (C) Brain activity in HC and PFC averaged across individuals.

and trends in information processing with disease progression. The modified model predicted cognitive trajectories with a performance comparable to other models presented in Table 1 (MAE = 0.539(0.087) and MSE = 0.786(0.218)) and demonstrated recovery/compensatory processes (Fig. 10).

## C.4 RL agent

The agent determines the change in information processing in each region. Our simulator consisting of DEs (Eqs. 1, 2, 3, 4, 5, 6, 8) allows us to use model-free RL with on-policy learning. During preliminary analysis (data not shown), we compared two of the most popular deep policy-gradient methods: TRPO [23] and PPO [45]. For both methods, we set the discount factor for generalized advantage estimates to be 0.97. For PPO, we clipped the likelihood ratio at 0.2 and used a minibatch size of 10. For TRPO, we imposed a constrain of 0.01 on the KL-divergence. We used the default values for the remaining hyperparameters. We compared the reward achieved by agents trained using TRPO and PPO and observed that TRPO outperformed PPO, for both synthetic and ADNI data (data now shown).

In our work, the policy agent is parameterized by a multilayer, feedforward neural network. We implemented the environment and the agent's interaction using OpenAI's Gym framework [24]. We used a stochastic two-hidden-layer Gaussian MLP with 32 neurons as the policy network, which we trained using the Garage Framework [46] (available under MIT License). During each epoch, the policy sampled 1000 trajectories from the simulator. An individual's trajectory consists of 11 time points including baseline. To stabilize learning, we imposed a minimum constraint of '-2000' on the reward and constrained the continuous action space in the range [-2,2]. The constraints on the action space were also motivated by the change in MMSE scores between subsequent years in ADNI data, $> 95\%$ of which lie in the range [-2, 2]. To constrain the agent from assigning $C(t) > 10$ to an individual, we incorporated a penalty factor in the reward function, based on the mismatch between $C_{task}$ and $C(t)$: $100^{[\max(C(t)-C_{task},0)]}$. Therefore, effectively, $R(t) = -[\lambda|C_{task} - C(t)| \times 100^{[\max(C(t)-C_{task},0)]} + M(t)]$. The model was trained on an internal multi-node compute cluster with two 20-core IBM POWER9 CPUs at 2.4GHz and 256 GB RAM.

The model requires $D(t)$ for the simulation although only $\phi(0)$ is available from baseline data. $D(1)$ is computed from $\phi(0)$ separately for each individual using the relationship between them provided in [5]:

$$\frac{d\phi(t_{po})}{dt_{po}} = \beta\tilde{H}(\beta t_{po})\phi(t_{po}),$$ (13)

$$\tilde{H}(\beta t_{po}) = U\,diag\left(\left\{\begin{array}{ll} \frac{1}{\beta t_{po}} & \text{if } j = 0 \\ \frac{\nu_j e^{-\nu_j \beta t_{po}}}{1 - e^{-\nu_j \beta t_{po}}} & \text{if } j > 0 \end{array}\right\}\right)U^T,$$

where $\nu_j$ are the eigenvalues and $U$ contains the corresponding eigenvectors of $H$, and $t_{po}$ denotes the time post onset of amyloid deposition. We approximate Eq. 13 to calculate $D(1) = \frac{\Delta\phi(t_{po})}{\Delta t_{po}} = \phi(t_{po} + 1) - \phi(t_{po})$, and set $\phi(t_{po}) = \phi(0)$ (amyloid deposition at baseline) and $t_{po} =$ individual's age at baseline $-50$.

We follow a two step procedure to determine $I(0)$ for an individual. First, we determine the best $I(0)$ for the entire population by using grid search as described in Section C.5. Second, we fine-tuned the $I^i(0)$ for an individual $i$ using the policy agent based on (i) their brain regions size $X^i(0)$, and (ii) the population $I(0)$ found from grid search. We observed that this two step procedure of selecting a population based $I(0)$ and then fine-tuning it for an individual allowed us to account for the variability in baseline cognition across individuals.

### C.5   Grid search for choosing $\lambda$ and $I(0)$

$\lambda$ and $I(0)$ must be specified for training the model and for predicting disease progression for an individual. We performed a grid search for $\lambda$ and $I(0)$ and chose the optimal value based on the average validation set error from cross validation. Grid search was performed independently for the synthetic data and for the ADNI data. We varied $\lambda \in \{2^{-1}, 2^0, 2^1, 2^2, 2^3\}$ for the synthetic data and $\lambda \in \{2^{-2}, 2^{-1}, 2^0, 2^1\}$ for the ADNI data based on preliminary analysis. $I(0)$ was chosen from the set $\{[10, 0], [9, 1], [8, 2], [7, 3], [6, 4]\}$ for both datasets, with the representation $I(0) = [I_{HC}(0), I_{PFC}(0)]$ for ADNI data. Based on the validation MSE (Table 3), we chose $\lambda = 2.0$ for synthetic data and $\lambda = 1.0$ for ADNI data. The optimal validation set error was achieved by $I(0) = [7, 3]$ for ADNI and $I(0) = [9, 1]$ for synthetic data (Table 3). The value of $\lambda$ and $I(0)$ identified during training was used while testing.

| MAE | ADNI | | | | Synthetic | | | | |
|---|---|---|---|---|---|---|---|---|---|
| | 0.25 | 0.5 | 1 | 2 | 0.5 | 1 | 2 | 4 | 8 |
| 6,4 | 7.730 (0.261) | 0.705 (0.180) | 0.499 (0.059) | 0.520 (0.093) | 2.230 (0.225) | 0.806 (0.069) | 0.711 (0.032) | 0.699 (0.116) | 0.754 (0.204) |
| 7,3 | 7.457 (0.255) | 0.724 (0.191) | **0.485 (0.087)** | 0.492 (0.091) | 2.292 (0.249) | 0.741 (0.115) | 0.766 (0.215) | 0.713 (0.084) | 0.700 (0.050) |
| 8,2 | 7.264 (0.274) | 0.732 (0.174) | 0.516 (0.072) | 0.495 (0.071) | 2.296 (0.094) | 0.764 (0.070) | 0.698 (0.112) | 0.739 (0.138) | 0.690 (0.163) |
| 9,1 | 6.805 (0.283) | 0.824 (0.178) | 0.529 (0.076) | 0.522 (0.093) | 2.166 (0.221) | 0.892 (0.206) | **0.670 (0.053)** | 0.781 (0.230) | 0.808 (0.228) |
| 10,0 | 6.348 (0.227) | 0.822 (0.346) | 0.527 (0.075) | 0.497 (0.099) | 2.057 (0.188) | 0.834 (0.130) | 0.904 (0.376) | 0.885 (0.321) | 0.891 (0.168) |

Table 3: Average validation MAE for different values of $\lambda$ and $I(0)$ for ADNI and synthetic data. Standard deviation is provided in parentheses. The lowest MAE for each dataset is highlighted.

## D   Benchmark and control models

### D.1   minimalRNN

Minimal recurrent neural network is a state-of-the-art model to predict cognition trajectories [10]. The model predicts AD progression up to 6 years into the future using data from one (baseline) or more time points. We used the open-source implementation of minimalRNN (GitHub link; available under MIT License). To maintain consistency of experiments across different models, only data at year 0 was input into the model. "Model filling" was used to infer values of brain region size ($X(t)$), amyloid pathology ($\phi(t)$), and age in subsequent years (see [10] for details). With the inferred $X(t)$, $\phi(t)$, and age along with other demographics (genetics and gender; $Z_0$), the minimalRNN model predicted cognition ($C(t)$) at each year. In this study, minimalRNN was trained for 100 epochs with a learning rate of $5 \times 10^{-4}$ and batch size of 128. The model is optimized by an Adam optimizer with weight decay set to $5 \times 10^{-7}$.

### D.2   SVR

Kernel support vector regression has been used as a benchmark model for cognition trajectory prediction [10]. It is a suitable learning based model in scenarios where the available data is limited. In our experimental settings, baseline brain region size ($X(0)$); amyloid pathology ($\phi(0)$); and demographics gender, genetic risk ($Z_0$), and age were input to the SVR model. SVR was implemented with the RBF kernel (with hyper-parameters: $C = 1$, $epsilon = 0.1$ and $gamma = 0.01$).

### D.3   Proposed model without RL

To assess the value of RL in our framework, we implemented our DE-based model and optimized $R(t)$ (Sec. C.4) at each time point using standard optimization instead of RL. Optimization of $R(t)$ was done independently at each time point. For each time point $t$, we performed a grid search by varying $I_v(t) \in \{0, 0.1, 0.2, ..., 10.4\}$ $\quad \forall v \in V$ and chose the value of $I(t)$ that maximized $R(t)$. $\lambda$

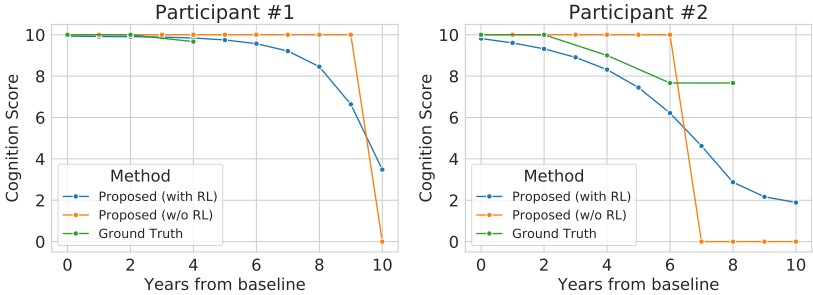

Figure 11: Cognition trajectories for two individuals. Ground truth trajectory and predicted trajectories (from proposed model with RL and proposed model without RL) are shown.

was the same as in the best performing proposed model reported in Table 1. The proposed model w/o RL as described above typically resulted in $I(t) \in \{[10, 0], [0, 10], [0, 0]\} \quad \forall t$ (Fig. 11).

### D.4 Control model for recovery

We implemented control models to evaluate the effect of the reward function on the recovery like behavior observed in Sec. 4. Specifically, we hypothesized that the shift in information processing observed in the proposed model was due to the form of the reward function $R(t)$ which consisted of a (i) cognitive mismatch term, and an (ii) energetic cost of cognition term. To test this hypothesis, we implemented two variants of the proposed model as control models with modified reward functions $R'(t) = -(C_{task} - C(t))$ and $R''(t) = M(t)$, respectively. The modified reward functions only consisted of one term each from $R(t)$. Note that, unlike $R(t)$ (Eq. 8), the modified reward functions did not need a $\lambda$ parameter. I(0) for the control model was the same as in the best model from Table 1. Information processing plots for the control models trained on ADNI data are shown in Figs. 12, 13.

The control model with reward $R'(t)$ roughly maintains the initial information processing distribution over regions, $I(0)$, throughout the entire 10 years (Fig. 12). Maintaining $I_1(t)$ at a high value leads to increased degeneration of the HC, which consequently increases $M(t)$. Since increased $M(t)$ does not penalize the reward, the control model approximately maintains $I(0)$ throughout. This suggests that $M(t)$ plays a role in demonstrating recovery in the proposed model.

The control model with reward $R''(t)$ pushes information processing in both regions to 0, i.e., $I(t) = [0, 0]$, $t \geq 1$ (Fig. 13). As expected, this model minimizes the energetic cost, i.e., $M(t) = 0$. This results in $C(t) = 0$, $t \geq 3$.

In summary, neither control model with the modified reward functions demonstrated recovery. Moreover, they highlighted the value of the two competing terms of (i) cognitive mismatch, and (ii) energetic cost in the reward function in demonstrating recovery in the proposed model.

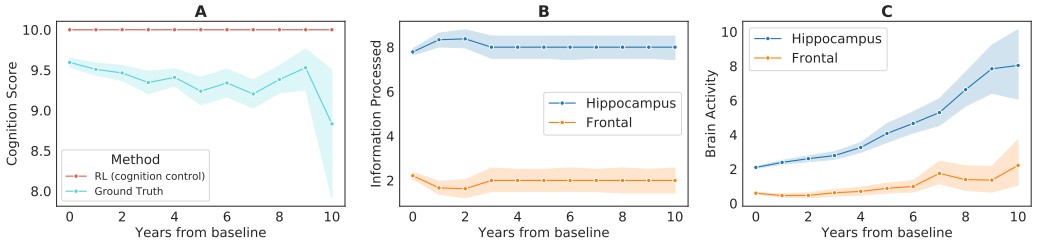

Figure 12: Control model with reward $R'(t)$ for ADNI. (A) Cognition trajectories from ground truth and model. (B) Information processed in HC (Hippocampus) and PFC (Frontal) averaged across individuals. (C) Brain activity in HC and PFC averaged across individuals.

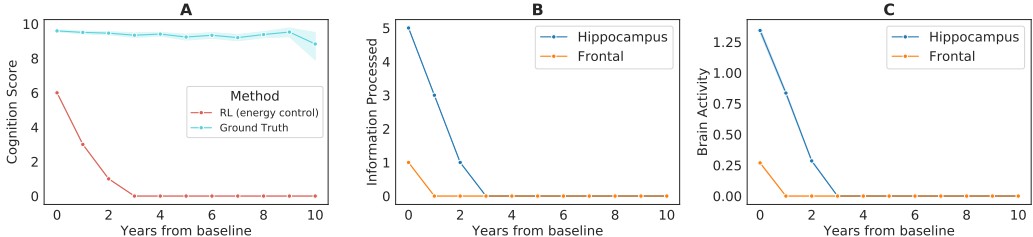

Figure 13: Control model with reward $R''(t)$ for ADNI. (A) Cognition trajectories from ground truth and model. (B) Information processed in HC (Hippocampus) and PFC (Frontal) averaged across individuals. (C) Brain activity in HC and PFC averaged across individuals.