# OpenReview forum: "Reinforcement Learning based Disease Progression Model for Alzheimer’s Disease"
_NeurIPS.cc/2021/Conference — NeurIPS 2021 Poster_

### Official Review · Reviewer_VwXC · 2021-07-15

**Rating:** 6
**Confidence:** 3

**Summary:**

The paper proposes a disease progression model for Alzheimer's disease (AD) that combines data-driven approaches capture trends in data over the course of the disease, which have proved successful over recent times, with more mechanistic biophysics-based models of how pathology spreads over the brain.  The authors introduce the use of reinforcement learning to learn update rules for regional brain quantities as a function of time thereby informing interactions between processes and measurements of neurodegeneration. Experiments with simulations confirm basic efficacy of the estimation procedure; experiments with ADNI data show some promising predictive power and demonstrate potential insights into disease biology.

**Ethical Concerns:**

Also not discussed, but again personally I don't see it as a problem.

**Limitations And Societal Impact:**

Not really discussed, although I don't see this as a concern for this work - it is well motivated.

**Main Review:**

This is a timely topic and extends the state of the art by bringing together data-driven models and mechanistic models.  The use of reinforcement learning is a nice idea, highly novel, and experiments show it makes a genuine impact on predictive ability.  The model overall ends up, as any model of such a complex process, making a lot of simplifying assumptions and so arguments in the paper about "explainability" in comparison to black-box fully data-driven techniques should be tempered by the fact that explainable observations are only meaningful if/when those assumptions hold. Nevertheless, the essence of the idea here extends as our understanding of AD improves and biophysical models become more accurate.  Overall this is a nice idea, although should be considered as a demonstrator of what's possible by fusing data-driven and mechanistic modelling approaches and avoid strong claims on what the model outputs actually say about AD; some of the results are anomalous to say the least, but the basic principles of the learning algorithms that enable such approaches are sound in my opinion.

Specific comments for the authors to consider:

1. Line 184 - I don't understand how the model can be estimated without functional MRI data informing on brain activation. It is unclear from the section that describes the data set, but this line suggests the paper uses no functional MRI at all; is that true and if so perhaps the authors could add some intuition to the formulation in Eq 9 explaining the consequences of not having this data available and how the I parameters, which depend on activation, can be defined without it.

2. Figure 4 desperately needs units to make any sense.  Particularly in panel A where the errors are all of order 10^-30 - I assume they are basically zero - but it depends on the scale of the measurements.  Does this suggest that the simulations use no noise perturbations of any kind so really that "validation" is simply a sanity check that the code works in estimating true parameters from data directly sampled from the model?

3. Section 4.2. can the authors confirm that the brain region sizes used to evaluate the model are not part of the training set to which the model is fit?  What is the ratio of hold-out data and how was the full data set divided into training/validation/test sets?

4. Figure 4C. The result that normal subjects appear to decline at the same rate as MCIs and much more quickly than SMCs is indeed highly anomalous - likely wrong.  This highlights some limitations in the model as it stands and I think it's important to get to the bottom of this kind of anomaly, although it doesn't detract from the potential of the learning algorithms that are the centrepiece of the paper.

5. The Introduction states  'we observed "recovery/compensatory" processes that mitigate the effect of the AD on cognition [11], which could not be observed with state-of-the-art models', which the qualitative discussion in section 4.4 reiterates.  It's unclear to me exactly what the authors mean by this - is it the opposing trends in hippocampal and frontal activity/processing-load that Figure 4B, 4C show?  The arguments that this genuinely reflects true trends in patients brains are very hand wavy and over-played in my opinion.  This either needs clarification or toning down substantially.

6. Line 295 - ref [28] doesn't appear to relate to event-based models - do you mean Fonteijn Neuroimage 2021?



**Time Spent Reviewing:**

2

---

> ### Author Response · Authors · 2021-08-10
> **response to Reviewer VwXC's comments**
>
> _"...how the model can be estimated without functional MRI data informing on brain activation...."_
>
> Functional MRI data was not used in this paper due to low availability in the cohort of 160 participants from ADNI that we studied. Due to space constraint, details of ADNI data are provided in the appendix (see B.2 lines 653-655). For estimating parameters $\alpha$’s and $\gamma$, which are affected by the unavailability of functional data ($Y(t)$), we eliminated $Y(t)$ from the equations by substituting Eq (4) in Eq (6), and then performed an L2 minimization using data and Eq (3). The intuition for the parameter estimation is provided in lines 185-187. Detailed derivation of the parameters is provided in the appendix A.4. One consequence of the unavailability of $Y(t)$ is that we use the same $\gamma$ for all the individuals (lines 191-194). Another is that a more refined version of Eq (4) could be developed if $Y(t)$ were available, which we have discussed in the limitations and future work (lines 337-340). As for the estimation of variable $I(t)$, we use RL for that. Essentially, the model estimates $I(t)$ from $X(t)$ (and $I(t-1)$) from the RL policy. Then, $I(t)$ and $X(t)$ are used to compute $Y(t)$ in the model. The computed value of $Y(t)$ along with information of amyloid deposition is used to update $X(t)$. This procedure is repeated to generate the cognition trajectory.
>
> _"Figure 4 desperately needs units to make any sense...."_
>
> Thanks for pointing this out! The parameters in the synthetic data are of the order of 10^-1 (appendix line 592), so 10^-30 is basically zero, as you pointed out. The simulation does not use noise perturbations in the parameters. We will modify Figure 4A to include the scales of parameter values in the synthetic data.
>
> _"Section 4.2. can the authors confirm that the brain region sizes used to evaluate the model are not part of the training set to which the model is fit...."_
>
> We confirm that brain sizes used to evaluate the model are not part of the training set for model fitting. We split the data into training-validation-test sets based on participants (ratio is 64:16:20) and performed a 5-fold cross-validation. Thus, in each fold, a participant appeared in only one of training, validation, or test set. Details of the cross-validation have been provided in Section 3.3 (lines 224-225).
>
> _"The result that normal subjects appear to decline at the same rate as MCIs and much more quickly than SMCs is indeed highly anomalous...."_
>
> Thank you! As mentioned in the manuscript (line 258), we agree that this result seems odd, but further analysis of the baseline characteristics of participants provides an explanation for this observation. We compared the baseline brain region sizes and amyloid deposition for the different diagnosis groups since baseline values strongly influence the predicted cognitive trajectories. Mean baseline brain region sizes for CN group were lower than SMC and more similar to MCI groups (mean region size for hippocampus- CN: 3.81, EMCI: 3.77, LMCI: 3.60, SMC: 3.94; PFC- CN: 3.71, EMCI: 3.81, LMCI: 3.74, SMC: 3.75). Similarly, mean baseline amyloid for CN participants was higher than SMC for hippocampus (CN: 1.32, EMCI: 1.29, LMCI: 1.29, SMC:1.28) although for the frontal areas was smaller than SMC (CN: 1.26, EMCI: 1.30, LMCI: 1.32, SMC: 1.33). Taken together, these observations suggest that the similar baseline brain region sizes and amyloid deposition of CN to MCI groups than SMC are the reason why predicted cognitive trajectories are similar for CN to MCI groups and worse than SMC. We will include this explanation and the supporting numbers and figures in the revised manuscript.
>
>
> _"....unclear to me exactly what the authors mean by this - is it the opposing trends in hippocampal and frontal activity/processing-load that Figure 4B..."_
>
> By compensation, we mean the opposing trends in information processing in Figure 5B that shows that as disease progresses, information processed in hippocampus decreases and in the frontal area increases (and then decreases after a certain point). This observation is consistent with biological studies that have proposed compensation, which is why we argue that the trends we observed could reflect trends in patients’ brains. Physiologic studies of functional brain changes in aging have consistently shown a posterior-anterior shift in aging (PASA) that has been interpreted as reflecting compensations [13]. This is also observed across the AD spectrum leading to the cascading network failure model of AD that’s also postulates a compensatory role for frontal hyperconnectivity in the setting of posterior hypoconnectivity that is related to the spatiotemporal evolution of amyloid and tau protein accumulation in AD [15] (Wiepert et al 2017). A quantitative comparison with those biological studies is not feasible due to differences in patient population and time scales of trajectories. We agree that the interpretation of these results is meaningful within the context of the assumptions of the model. We will clarify the above in the revised manuscript.
>
>
> Wiepert, Daniela A., et al. "A robust biomarker of large-scale network failure in Alzheimer's disease." Alzheimer's & Dementia: Diagnosis, Assessment & Disease Monitoring 6 (2017): 152-161.
>
> _"...ref [28] doesn't appear to relate to event-based models..."_
>
> Thanks for catching that! Indeed, we were referring to the event-based model paper from Fonteijn et al. in NeuroImage. We’ll fix the reference in the revised manuscript.

---

### Official Review · Reviewer_31zC · 2021-07-15

**Rating:** 7
**Confidence:** 3

**Summary:**

This work presents a method for modeling dynamic systems that combines mechanistically motivated differential equations with data driven models learned by RL.  The method is applied to AD progression, measured by MMSE (a cognitive function test) scores.  The key idea is that a mechanistic understanding of the system under study may be incomplete -- e.g., in a system of coupled differential equations, a quantity and its time evolution may be unspecified -- but that it may be possible to fill this gap by learning a reasonable model for the missing function.  This method is applied to AD progression in a real world dataset with mixed results - although adding the learned dynamics model improves accuracy (measured by MSE) a great deal relative to the DE baseline, it does not do as well as a pure ML based baseline (SVR).

**Limitations And Societal Impact:**

The authors point out one limitation - only 2 brain regions were modeled.  I don't think this is a very big deal.


**Main Review:**

This work presents a fascinating approach to modeling dynamical systems in which there is a partial mechanistic understanding of the dynamics, enabling the specification (and fitting from data) of a detailed but incomplete system of differential equations.  Specifically, let $v$ index brain regions.  They hypothesize a mechanistic model of AD progression involving brain region size $X_v$, regional activity $Y_v$ (information processing per unit size), and a $I_v$ representing total information processing in each region.  Note that $Y_v X_v = \gamma I_v$ - $X_v$ can be directly observed from brain scans, as can $Y_v$ (via functional MRI). The hypothesis further specifies - and this is the crux - the time evolution of $X_t$: it is thought that degeneration of a region $v$ (i.e., changes in size) is governed by amyloid deposition and activity $Y_v$, with patient specific modifiers.  This is formulated as  a set of differential equations.  But these differential equations are incomplete - they specify the time derivatives of $X_v(t)$, but not $Y_v(t)$, or equivalently, $I_v(t)$.  This is analogous to trying to solve a linear system in 3 unknowns from only 2 equations.  So if we want to, e.g., run the system forward from a set of initial conditions, we are stuck.

How can we fill in the missing piece, I(t)? Rather than hypothesize a differential equation for $I_v(t)$, this work proposes using RL to learn a policy that optimizes an intuitively reasonable objective - you want to increase $\sum_v I_v(t)$ to meet cognitive demands but also want to decrease metabolic demands. The action space consists of setting of I_v(t) to meet cognitive demand (minimized $C - \sum_v I_v(t)$) while decreasing metabolic demand $\sum_v Y_v(t) = \sum \frac{I_v(t)}{X_v(t)}$.  The dynamics of the RL system are determined by the system of equations, with parameters fit to training data (with some caveats due to the fact some patients do not have complete measurements for $Y_v(t)$) - the key link is that increased activity can lead to degeneration (negative changes in $X_v$).  Now we can start from initial conditions, say year 0 of AD, and then run the system forward in time using the learned policy in place of a discretized differential equation for $I$ to forecast progression, and compare to the observed trajectories and other forecasting methods.  In this paper, they do this for a 10 year window, with one year time steps.  There is an evaluation on synthetic data, but I will focus on the evaluation on the real data (ADNI).  I am not familiar with the dataset used.

How did they do?  There are two comparisons. First, is the usual bake off comparison of their method vs some solid baselines.The new method seems to do very well wrt MSE compared to one baseline, and about the same as another baseline, SVR, in terms of agreement with the observed progression of test patients.  Plotting the data reveals a nuance, however - the SVR baseline fits the observed data shape, which shows a weird increase in mean cognitive function in year 9 followed by a precipitous decline in year 10 - better than the new method, which forecasts (on average) a smoother, monotonic decrease in cognitive function.  The authors explain this away, asserting that the smoother forecast is more realistic.  I agree, and am willing to view this as a case where noisy data is mitigated by a (semi-)mechanistic formulation of the dynamic model. In another context, the authors also raise the issue of missing measurements in the (real) dataset - since it is so sparse (31% observed) it may be that this is a result of an unfortunate pattern of missingness towards the end of the observation period.  This could perhaps be checked (are the patients who are contributing to the population mean at 9 years the same patients contributing to the population mean at 8 years, etc).

The second comparison is a bit more puzzling - it shows that the forecasts with and without RL are remarkably similar in shape, with a scaling or constant offset from each other over time.  This suggests that the RL component is mostly setting an offset/constant - this is perhaps not surprising given the form of the differential equations, etc.  But it does beg the question of whether there might have been a simpler way to arrive at a dynamical system that fits the data very well (e.g., fitting a constant relating the initial year 0 prediction to observed on the training data, i.o.w., fitting an intercept), and whether the RL component is learning a trivial policy.

There is an auxiliary result - their learned dynamical system also demonstrates compensation, a well known phenomenon in which cognitive demands can be shunted around to different regions to compensate for degradation of some areas.  This seems to me to be an unsurprising result of the RL objective, so is not that noteworthy.

Overall I liked this work very much as a demonstration of a cool idea I haven't seen elsewhere.  There are elements of the presentation that should be clarified, along with some additional questions that it would be nice to see addressed.

First, it is not clear to me how the baseline without RL is accomplished.  The whole point was, I thought, that we couldn't run the dynamical system forward from initial conditions because we are missing a key piece we will fill in with a learned model.  So how was the baseline run?

Second, how did the authors handle the amyloid deposition as a function of time, $D_v(t)$, introduced in equation 6, line 142?  It also seems unspecified in the dynamical system - don't you have to model that somehow too?

Third, in figure 5A, it would good to know why, at t = 0 years, most of the estimates are offset from the ground truth mean.  I would have thought t = 0 corresponds to the initial conditions, so what does a forecast from a dynamical system mean at that time?

Fourth, I am at a bit of a loss as to how this could be done, but I do wonder whether the RL policy governing $I_v(t)$ is accurate and "non-trivial".  This is not possible with the real dataset, and I doubt there are any datasets of fMRI over 10 years of AD progression, but this does seem to me to be an important point about whether this approach is really adding a lot.  Perhaps in the synthetic data the authors could directly evaluate the RL policy governing the time evolution of $I_v(t)$, both when the data generating process is a good fit for the hypothesis, and when it is not?

Finally, if possible, it would be interesting to see a sensitivity analysis to see how robust results are to misspecification of both the differential equations and the learned model for $I_v(t)$.  In general, I suspect that this could be a big limiter to this general approach - we may have only a very vague sense of the mechanisms (leading to misspecification of the dynamical system) and how to formulate a good RL objective.

Overall, I really enjoyed this work - it was quite different from the usual ML conference submission and I think it could spark a lot of cool work and discussion even if it doesn't convincingly, in my opinion, beat one of the baselines or work super well.  It's hard to say how generally applicable this approach could be since it does require a lot of (reliable) domain knowledge, and it's not clear to me how well it worked in this case, but I found the approach intriguing and worth introducing to the NeuRIPS community.  I am curious whether this sort of approach has been taken in, e.g., applications of ML to physics.



**Time Spent Reviewing:**

5

---

> ### Author Response · Authors · 2021-08-10
> **response to Reviewer 31zC's comments**
>
> _"...perhaps be checked (are the patients who are contributing to the population mean at 9 years the same patients contributing to the population mean at 8 years..."_
>
> Thank you for the detailed reviews and the comment on the novelty of the work! As pointed out in the manuscript (line 271), 17 participants have data for year 9 and 4 for year 10. There are 60 participants with cognitive score at year 8 out of which 13 have data at year 9. The unavailability of contiguous data from each participant further supports the assertion that missingness in the data plays a strong role in the trends learned by the SVR model.
>
>
> _"....it shows that the forecasts with and without RL are remarkably similar in shape..."_
>
> This is a keen observation based on the average trends in Fig 5A! When considering individual participant’s trajectories, the model without RL resulted in cognition being either 10 or 0 at each time point, whereas the output of the RL model provided more “realistic” trajectories for individuals. We have discussed about the behaviour of the model without RL in Appendix D.3 (line 748). Essentially, the model without RL is solving a linear optimization at each time point independent of the future or past time points, resulting in $I(t)$ taking on the form (10,0), (0,10), or (0,0). To demonstrate that the model with RL is providing non-trivial trajectories compared to the model without RL, we will provide sample trajectories for some individuals in the revised manuscript.
>
>
>  "... seems to me to be an unsurprising result of the RL objective..."_
>
> We agree that the compensation behaviour is a result of the RL objective, as also supported by the control experiments in which we modified the RL objective function and did not observe compensation (appendix D.4). This result also provides a validation of our model.
>
> _"First, it is not clear to me how the baseline without RL is accomplished..."_
>
> The model without RL was accomplished by optimizing for $I_v$ for each time point, independent of other time points. The same objective function R(t) was used at each time point. We used grid-search to find the optimal solution. Details of these are presented in Appendix D.3. A grid-search is possible because the reward is optimized independently for each time point, resulting in non-realistic trajectories for each individual (cognition either 10 or 0 at each point). We will add a brief comment on the procedure for optimization without RL in the manuscript to enhance readability.
>
> _"Second, how did the authors handle the amyloid deposition as a function of time..."_
>
> We modelled the amyloid deposition based on a differential equation model previously proposed in literature. The differential equations for it are provided in Eq (1) and (2) on lines 116 and 118. Details of the initialization in the dynamical model based on real data are provided in Appendix C.3 (lines 700-706).
>
>
> _"Third, in figure 5A... at t = 0 years, most of the estimates are offset from the ground truth mean."_
>
> Thanks for the keen observation! The model takes as input the baseline demographics, brain regions, and amyloid deposition to predict the rest of the variables. Since I_v(0) is unavailable in the data, we estimate it from the input features. Thus, the model estimates the cognition at time 0 based on the input features instead of using the baseline cognition value, due to which there is an offset. Description of the estimation procedure of baseline cognition is provided in the appendix C.3 (lines 707-712).
>
>
> _"Fourth...whether the RL policy governing $I_v(t)$ is accurate and "non-trivial"...."_
>
>
> We visually assessed the trajectories of several individuals and found that the trajectories varied based on the input features. Such variation is observed in the synthetic data as well as the ADNI data. We will provide trajectories for some individuals in the appendix to provide support to the argument that the RL policy is non-trivial. Further evidence is provided by the two controls models from Appendix D.4 and in the proposed model trained without RL. In the control models, we modified the objective function to have only one term (maximizing cognition or minimizing cost), and observed that the learned RL policy was markedly different from the proposed model (as seen by the average cognition trajectories in Appendix Fig. 9 and 10). Specifically, we observed that both those terms in the reward function (Eq (8)) were needed to obtain realistic predicted cognition trajectories. Finally, as mentioned above, training the model without RL resulted in cognition trajectories with cognitive score of either 10 or 0.
>
>
> _"...it would be interesting to see a sensitivity analysis...."_
>
> Thanks for the great point! Our analyses so far supports the suggestion that the form of the differential equation matters in the model’s prediction ability, but the extent of the effect is unclear. For example, the formulation of the RL objective function significantly impacts performance (see appendix D.4). We also tested an alternate model where the differential equations corresponding to amyloid deposition and degeneration were modified (see appendix C.2). This alternate model could not be tested due to limited data. Finally, in another set of experiments, we modified the form of Eq (4) (line 131) to have a I_v/X_v^2 relation and evaluated the modified model’s prediction ability. In those experiments, we observed that this modified model had prediction MSE/MAE values similar to the ones in Table 1. The results for this specific modification will be added to the revised manuscript. We will add a comment relating to the sensitivity to differential equation specification in the conclusion of the revised manuscript.
>
> _"...I am curious whether this sort of approach has been taken in, e.g., applications of ML to physics."_
>
> Previous studies in application of ML to physics have used differential equations. Though not directly related to our manuscript, techniques have shown how deep learning can be applied to enable the search for ODEs/PDEs in physics (e.g., Raissi 2018 for fluid dynamics). On the other hand, domain knowledge from physics (in the form of ODEs) has been applied as regularizes for AI models (e.g., Stewart and Ermon 2017).
>
> Raissi, M. (2018). Deep hidden physics models: Deep learning of nonlinear partial differential equations. The Journal of Machine Learning Research, 19(1), 932-955.
>
> Stewart, R., & Ermon, S. (2017, February). Label-free supervision of neural networks with physics and domain knowledge. In Thirty-First AAAI Conference on Artificial Intelligence.

---

### Official Review · Reviewer_zqhh · 2021-07-16

**Rating:** 5
**Confidence:** 3

**Summary:**

The authors proposed a method that combines differential equation modeling with reinforcement learning to predict the progression of Alzheimer’s disease.

In terms of contributions, the authors claimed that this study is the first to use reinforcement learning for disease progression. During my non-exhaustive literature review there does not seem to be clear objections against this statement. The authors also claimed that their model showed a 11% lower prediction error compared to a state-of-the-art. Moreover, the authors mentioned the so-called “recovery/compensatory processes” that is present in Alzheimer’s disease could be observed in their proposed model, showcasing the interpretability of the model.

**Limitations And Societal Impact:**

Nothing seems to be concerning.

**Main Review:**

Strengths:
1.	Personally, I am in favor of the approach the authors take, where they use explicit equations instead of an arbitrary machine learning model to represent processes and behaviors that are relatively well-studied. In this way they utilized domain knowledge in an intuitive and explainable manner.
2.	The authors tend to think proactively about interpretability of their methods, and they related their observations with the “recovery/compensatory” process known in neurobiology.

Weaknesses:
1.	In introduction, the authors mentioned that existing mathematical models cannot faithfully represent all factors relevant to Alzheimer’s disease. Based on that, it would be reasonable to either refine the equations or use a different model to take care of the gap. However, it is not very intuitive why reinforcement learning is a natural choice. It would be better if some insights and rationale is given.

Additional feedback:
1.	In Figure 4B, it might be helpful to indicate the correlation coefficients, even though it is mentioned in the text. Also, it would make more sense if the predicted vs. ground truth size of PFC is included – it looks weird when both HC and PFC are described in the text but only HC is shown in the corresponding figure.
2.	Captions of Figure 5 needs a bit improvement. It is not clear which model variant Figure 5B and 5C correspond to.
It would be interesting to see whether the model can perform consistently when choosing different follow-up times (year-1, year-2, etc.) as the initial point. For example, the prediction of N years after year-0 shall be reasonably similar to that of N-1 years after year-1. At the same time, I am aware that some data might only be available at year-0 and this might not be an easy investigation.

**Time Spent Reviewing:**

8

---

> ### Author Response · Authors · 2021-08-10
> **response to Reviewer zqhh's comments**
>
> _"...it is not very intuitive why reinforcement learning is a natural choice."_
>
> Thank you for the comment! We have DEs but they only explain a part of disease progression. Refining those DEs requires significant scientific knowledge and that knowledge is still in development (as we see it) for AD progression. An alternative approach would be to fill the gap with data and the available domain knowledge -- RL offers a method for doing the same. While there may be other approaches that could follow ours, RL offers the best opportunity to address the gap today.
>
> To complete the model, we need a relationship to explain how information processing is distributed across brain regions. Intuitively, the distribution of information processing depends on the cost of cognition and the cognitive demand. Those requirements can be converted into an optimization problem using the reward function (Eq (8)). Moreover, the optimization is over a sequence of states (trajectory) since the information processing at one time point affects the brain region sizes at future time points. RL provides a method for solving this optimization problem.
>
>
> _"In Figure 4B, it might be helpful to indicate the correlation coefficients..."_
>
> We will add the plot for PFC and the correlation coefficient on the plot in the revised manuscript.
>
> _"Captions of Figure 5 needs a bit improvement...."_
>
> Thank you for pointing this out! 5B and 5C correspond to the proposed model with RL. We will clarify that in the revision.
>
> The consistency of predicted trajectories for different follow-up times is a very interesting suggestion! As the reviewer rightly pointed out, the main limiting factor for this analysis is the (un)availability of data at follow up times. None of the participants have both MRI and amyloid measurements at year-1, due to which we cannot compare the performance by choosing year-1 as “baseline”. 134 participants have all the relevant data available at year-2. For those participants, we predicted trajectories using year-2 as “baseline” and compared them with the trajectories using year-0 as baseline. The two trajectories were highly correlated across participants (correlation=0.93). We did not compare trajectories using year-3 as baseline since only 3 participants have measurements of both MRI and amyloid on that visit. We will include this analysis on the consistency of predicted trajectories in the revised manuscript.

---

### Official Review · Reviewer_2T27 · 2021-07-21

**Rating:** 6
**Confidence:** 4

**Summary:**


This paper addresses the problem of predicting long-term Alzheimer’s disease (AD) progression. It proposes a model that combines domain knowledge on the evolving relationships between factors affecting the brain together with a data-driven approach to learning the missing relationships. The learning objective used to learn the data-driven components satisfy a general criteria about the working of the brain, which maximizing cognition while minimizing the cost of supporting cognition. The authors propose using differential equations (DEs) to capture domain knowledge and reinforcement learning (RL) to learn the data-driven component. The model was validated on the ADNI data set and was found to outperform SOTA baselines.

**Limitations And Societal Impact:**

Limitations were addressed. No negative social impact.

**Main Review:**


This paper addresses a very important disease that presents one of the biggest emotional and financial burdens to patient care takers and healthcare systems, and presents an interesting approach for combining domain knowledge and data-driven model components. However, the model was only evaluated on a very small sample size (160 patients from the ADNI data set) and did not consider a lot of previous work that developed disease progression models for the same disease and the same data set. Because of these reasons, I think the paper belongs in the **borderline acceptance** category.

Few comments and questions that I'd like the author to reply to:

- It is unclear to me what is the reasoning behind using RL to solve the optimization problem for $I(t)$. Since we collect all the data retrospectively and run the model on an offline, doesn't it make more sense to solve the optimization problem for all $t$ using an offline approach?

- What is the clinical significance of the results in Table 1? The MSE and MAE by themselves do not reflect accuracy of predicting clinically-useful phenotypes of dementia. Did you consider predicting discrete 10-year outcomes (e.g. conversion of mild cognitive impairment to AD) and evaluate the model accuracy instead?


**Time Spent Reviewing:**

5+ hours

---

> ### Author Response · Authors · 2021-08-10
> **response to Reviewer 2T27's comments**
>
> _"...the model was only evaluated on a very small sample size (160 patients from the ADNI data set) and did not consider a lot of previous work that developed disease progression models for the same disease and the same data set."_
>
> Thank you for the comments! The reason we used a small subset of the ADNI data was because of the stringent inclusion criteria for the participants in this study (see Appendix B.2 lines 628-632) – only 160 participants had sufficient data (at least 3 time points of concurrent MRI and amyloid data) to support parameter estimation. In future work, we plan to access a larger database and relax the inclusion criteria for individuals for the analyses.
>
> We consider our treatment of the previous work well representative of the work in the related area. A detailed review of the relevant studies is provided in Related Work (Section 5). Several of those studies only look at short term predictions (< 5 years), so are unsuitable for comparison. We compared with minRNN, which is among the top performing models in the TADPOLE challenge and predicted long-term cognition trajectories, and found our model to be 11% better in terms of MSE. We are unaware of any fully mechanistic models for AD progression for comparison.
>
>
>
> _"... the reasoning behind using RL to solve the optimization problem for $I(t)$."_
>
> RL is a desirable method in our case since it can train with limited data and makes the framework generic and flexible to changes/refinements in the reward function and the underlying differential equations (DEs). In Table 1, we have compared with “offline” learning techniques and our model performs favorably.
>
> Based on the reviewer’s comment, we explored techniques that use offline optimization along with the DE constraints. We identified optimal control-based approaches like the one presented by Culshaw et al (2004). However, these techniques are not as flexible as our RL method since their solution equations need to be re-derived if the model DEs and/or the cost (reward) function is changed (which is challenging and can potentially lead to requiring numerical solvers). Moreover, offline approaches can be data hungry, thereby resulting in poor performance (in terms of MSE/MAE) with the limited available data. Since we create a simulator with the DEs, RL can train with the available data.
>
> Culshaw, Rebecca V., Shigui Ruan, and Raymond J. Spiteri. "Optimal HIV treatment by maximising immune response." Journal of mathematical biology 48.5 (2004): 545-562.
>
>
>
> _"What is the clinical significance of the results in Table 1?"_
>
> We agree that predicting conversion from MCI to dementia is clinically meaningful, but it is also very meaningful to predict cognitive trajectories captured by our model. These trajectories vary considerably between patients making counseling, management, and clinical trial design difficult (Jutten et al 2021). In our discussions with neurologists, we see a hope that our model will improve this situation while also informing the underlying physiological mechanism for the observed heterogeneity in cognitive trajectories. Therefore, the improved prediction accuracy of our model compared to a SOTA model is significant. Moreover, evaluating the prediction accuracy of discrete outcomes is difficult for the cohort of subjects included in this study since 10-year diagnosis data is available only for a handful of participants (4 participants at year 10, 7 participants have diagnosis data for 9 years).
>
> Jutten, Roos J., et al. "Finding treatment effects in Alzheimer trials in the face of disease progression heterogeneity." Neurology 96.22 (2021): e2673-e2684.

---

### Comment · Area_Chair_SgQ8 · 2021-08-19
**Discussion**

I wish to thank the reviewers and authors for the extremely thorough reviews and rebuttals provided. Given the author's comments, I would like to invite the reviewers to weigh in on the discussion: do you think that the authors have adequately addressed your comments? Do you think that the comments of the other reviewers have been addressed? Given the rebuttal, would you be willing to update your score? Given the variance of the scores, it would be good to see if we can converge towards a consensus. This makes it important for us to start a discussion around this.

---

> ### Comment · Reviewer_2T27 · 2021-08-26
> **My score is unchanged, leaning towards acceptance**
>
> Having read the author responses and the other reviews, I don't think that my overall evaluation has changed. Overall, the paper addresses a very important clinical problem, even though the technical contribution and the size of the validation data are limited. I think the proposed approach for combining mechanistic and RL-based data driven modeling approaches is interesting and worthy of publication.

---

### Decision · Program_Chairs · 2021-09-27

**Decision:**

Accept (Poster)

**Comment:**

In the discussion with reviewers, the reviewers have highlighted the merits of this paper. One particular reviewer was unable to update their score of 5 to an accept, however, in private conversation with the reviewers, they all agreed that this paper warrants acceptance. This paper addresses a very important clinical problem, even though the technical contribution and the size of the validation data are limited. The proposed approach for combining mechanistic and RL-based data driven modelling approaches is interesting and worthy of publication.